# CL-GCL: Comprehensive and Lightweight Graph Contrastive Learning

**Jianqing Liang** [1]  **Xinkai Wei** [2]  **Zhiqiang Li** [2]

## Abstract

Graph Contrastive Learning (GCL) has significantly advanced self-supervised representation learning on graphs, yet its practical efficacy remains hindered by random augmentations that induce semantic distortion and rigid one-to-one sampling strategy that amplifies inter-class entanglement and intra-class dispersion. To address these limitations, we develop CL-GCL, a Comprehensive and Lightweight Graph Contrastive Learning framework. Specifically, we exploit graph coarsening to preserve structural semantics through community-level representations and manifold learning to capture local geometric relations without costly pairwise distance computations. This design naturally aligns with the neighborhood aggregation principle of Graph Convolutional Networks, enhancing structural consistency while eliminating negative sampling bias. We theoretically prove that CL-GCL approximates node-level contrastive loss under mild conditions. Extensive experiments demonstrate consistent superiority in both accuracy and efficiency over state-of-the-art GCL methods.

## 1. Introduction

Graph Neural Networks (GNNs) are powerful graph representation learning architectures, which evolve significantly across diverse domains, including molecular property prediction (Liang et al., 2024), social networks (Pal et al., 2020; Yang et al., 2021) and point clouds (Du et al., 2024). However, conventional GNNs (Kipf & Welling, 2017) typically require extensive manually labeled data, which are expensive and labor-intensive to collect. A promising line of recent studies has incorporated self-supervised learning (SSL) into graph representation learning, to address label sparsity issue with self-supervision signals. Contrastive learning, as a dominant SSL paradigm, has demonstrated impressive progress in the field of computer vision (Chen et al., 2020; Grill et al., 2020). In light of the success, graph contrastive learning (GCL) has drawn recent interest for learning generalizable and robust representations from unlabeled graph data (Velickovic et al., 2019; Zhu et al., 2020; 2021; You et al., 2021; Xia et al., 2022b; Niu et al., 2024). Among all these method variants, effective graph augmentation and node sampling strategies are essential for GCL to achieve success (Suresh et al., 2021; Wei et al., 2023).

For conventional GCL, random augmentation is still a default (You et al., 2020; Qiu et al., 2020; Zhu et al., 2021; Zhang et al., 2023; 2022). Typical random augmentation strategies include node dropout (You et al., 2020), edge perturbation (Qiu et al., 2020; Zhang et al., 2023) and attribute masking (Zhu et al., 2021; Zhang et al., 2022). GRACE (Zhu et al., 2020) utilizes random topology perturbation and node attribute masking to generate two augmented views. Similarly, BGRL (**?**) and ProGCL (Xia et al., 2022b) rely heavily on random augmentations to generate multiple views. In spite of the performance comparable to supervised learning methods, several intrinsic drawbacks such as semantic information corruption still remain to be solved. Several works also seek to integrate expert knowledge into graph augmentation frameworks. GCA (Zhu et al., 2021) employs network science principles to regulate edge removal probabilities. Nevertheless, these random or knowledge-driven augmentation approaches demonstrate dataset-dependent sensitivity (Shen et al., 2023) and can result in performance degradation (Yin et al., 2022). To enhance generalization capability and achieve global optimization, researchers introduce learnable graph augmentation techniques that systematically eliminate redundant information while preserving essential features across augmented views (Tong et al., 2021; Suresh et al., 2021). Despite these advancements, current random graph perturbation in augmentation inevitably corrupts high-level graph information such as community, which is fundamental for downstream applications such as node classification and link prediction (Li et al., 2022a; Chen et al., 2023).

Current node sampling strategies primarily focus either on topology structures or node features (Li et al., 2022b). The

---

[1]School of Computer Science and Engineering, Southeast University, Nanjing 210096, Jiangsu, China [2]School of Computer and Information Technology, Shanxi University, Taiyuan 030006, Shanxi, China. Correspondence to: Jianqing Liang <liangjq@seu.edu.cn>.

*Proceedings of the $43^{rd}$ International Conference on Machine Learning*, Seoul, South Korea. PMLR 306, 2026. Copyright 2026 by the author(s).

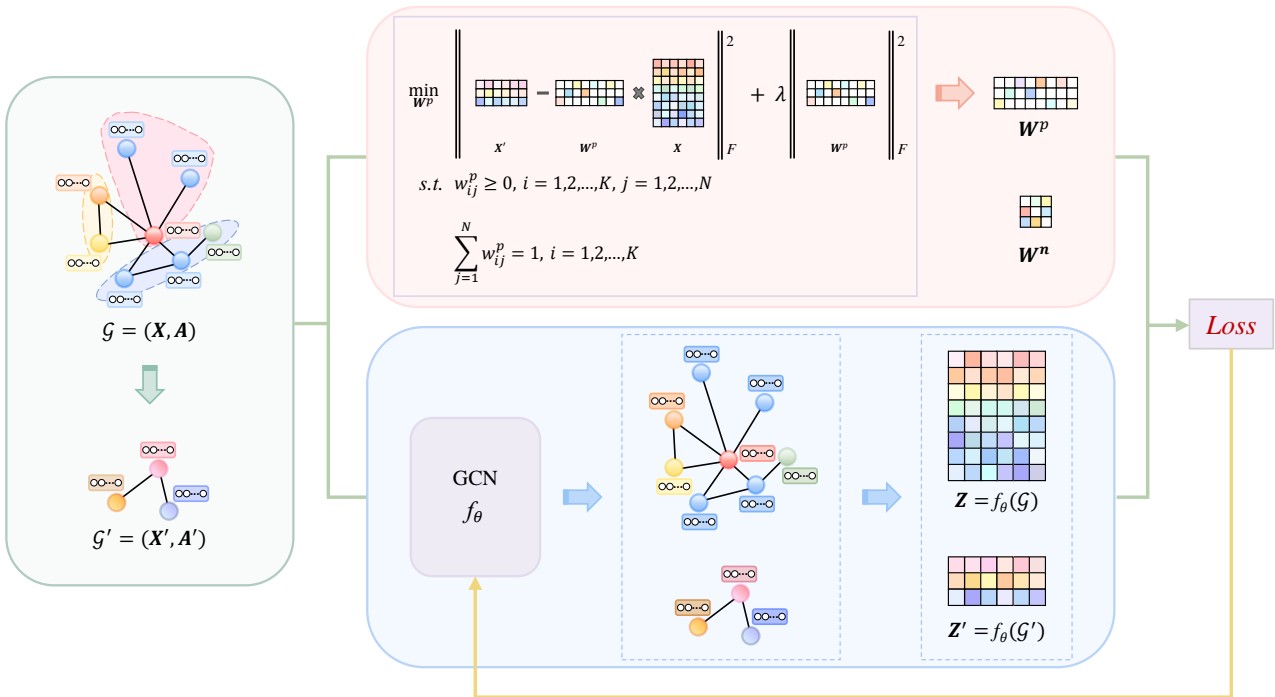

*Figure 1.* The overview of CL-GCL. With original graph $\mathcal{G}$ and coarsened graph $\mathcal{G}'$, we compute the positive pairs weight matrix $\boldsymbol{W}^p$ with a closed-form solution of a manifold learning inspired optimization problem. Then, we calculate the negative pairs weight matrix $\boldsymbol{W}^n$. Next, we apply GCN as a graph encoder to obtain the node embeddings $\boldsymbol{Z}$ and $\boldsymbol{Z}'$. Last, we compute the normalized loss function with regard to the positive pairs weight matrix $\boldsymbol{W}^p$, negative pairs weight matrix $\boldsymbol{W}^n$ as well as the node embeddings $\boldsymbol{Z}, \boldsymbol{Z}'$ and then update the graph encoder.

one-to-one sampling strategy (Zhu et al., 2020; 2021) follow the rigorous paradigm that nodes with the same index of different augmented views are regarded as positive pairs. Such strategy builds upon the label consistency assumption (Wang et al., 2022b) that data augmentations will not alter semantic information, which is almost impossible for GCL. In computer vision, randomly cropping or changing the color of an image, its underlying semantic still remains the same. However, in biochemistry, removal of an atom or edge from the molecules may change their structures, resulting in different compounds. In light of these risks, subsequent works attempt to incorporate additional structural and semantic information to improve sampling quality. For example, HomoGCL (Li et al., 2023b) uses a Gaussian Mixture Model (GMM) to perform soft clustering over nodes, estimating the probability of neighbor nodes being positives. HEATS (Zhuo et al., 2024) performs adaptive positive sampling using an affinity matrix with a block-diagonal structure and idempotent constraint. GTCA (Liang et al., 2025b) constructs positive pairs with the intersection $k$-NN of the anchor node with GCN, NodeFormer encoders and topology structure. While these methods enhance the semantic relevance of sampling to some extent, they still rely heavily on heuristic rules, which can introduce false positives. Moreover, their dependence on negative pairs remains substantial, ultimately limiting their applicability to large-scale graphs.

In this paper, we propose a Comprehensive and Lightweight Graph Contrastive Learning (CL-GCL) framework. As Figure 1 presents, CL-GCL introduces graph coarsening to extract critical structures from the original graph and constructs contrastive pairs on the community level. Specifically, nodes within the same community are treated as positive pairs, while nodes across different communities form negative pairs. This strategy significantly reduces the number of negative pairs and lowers computational costs, thereby improving the scalability on large-scale graphs. Furthermore, we introduce a local linear fitting strategy to approximate the graph's global nonlinear structure, implementing a soft selection mechanism within the community, thereby effectively alleviating sampling errors. Finally, we design a novel community-node mixed contrastive loss to improve the robustness and discriminative power. Our main contributions are as follows:

- We propose a CL-GCL framework to achieve critical

structure preservation, global nonlinear recovery and computational efficiency optimization.

- Theoretical analysis offer insight into how close CL-GCL is to the contrastive loss on the original graph. Specifically, we give and prove this gap theoretically.

- Extensive experiments demonstrate the effectiveness and efficiency of CL-GCL across multiple graph datasets, yet without resorting to domain knowledge.

## 2. Related Work

### 2.1. Graph Contrastive Learning

In the past decades, contrastive learning has advanced graph representation learning paradigm with unlabeled graph data (Velickovic et al., 2019; Hassani & Khasahmadi, 2020; Peng et al., 2020; Zhu et al., 2020). GraphCL (You et al., 2020) generates correlated graph representation views with various augmentation strategies, such as node/edge perturbation and attribute masking. GCA (Zhu et al., 2021) performs attribute-level and topology-level data augmentation for multiple views generation. In this way, crucial edges and features can be identified. SimGRACE (Xia et al., 2022a) utilizes GNN encoder perturbations to generate contrastive views. While these methods have achieved significant success, such random augmentation may disrupt underlying semantics, and lead to substantial computational overhead.

### 2.2. Node Sampling

According to node sampling strategies, current GCL methods can be classified into local-local (Zhu et al., 2021), global-local (Velickovic et al., 2019) and global-global (You et al., 2020) modes.

Local-local contrasting mode selects positive/negative pairs on node level. GRACE (Zhu et al., 2020) treats the same nodes from two augmented graphs as positive pairs and all the other nodes as negative pairs. GCA (Zhu et al., 2021) further equips GRACE with adaptive data augmentation, learning important patterns underneath the input graph. It is noteworthy that these methods regard neighbors as negatives and push them far away from the anchor. Such node sampling strategy contradicts the homophily assumption of GNN that connected nodes often belong to the same class. NCLA (Shen et al., 2023) regards intra-view and inter-view neighbors as positives. HomoGCL (Li et al., 2023b) applies Gaussian Mixture Model (GMM) to obtain soft clustering assignments, where node similarities are computed as the indicator of the positive pairs probability. Recently, GTCA (Liang et al., 2025b) inherits the advantages of both GNN and Transformer, incorporating graph topology to obtain comprehensive graph representations. While some progress has been made, a large number of pairs may lead to high computational costs.

Global-global and global-local contrasting modes define positive/negative pairs on graph level. DGI (Velickovic et al., 2019) and MVGRL (Hassani & Khasahmadi, 2020) maximize the mutual information between the node and the global summary of the graph to learn node representations. In spite of the competitive performance, global features may be insufficient to retain node-level embedding information. GraphCL (You et al., 2020) takes multiple augmented graphs of original graph as positives, and other graphs in the same mini-batches are negatives. However, augmentation randomness inevitably leads to undesired semantic loss. GCS (Wei et al., 2023) can effectively help disentangle semantic and environmental background structures with gradient-based saliency measurements, thereby constructing positive pairs that preserve the semantics within the graph.

### 2.3. Scalability

The scalability of GCL is a critical challenge in handling large-scale graphs. Most of existing methods, such as GraphCL (You et al., 2020) and MVGRL (Hassani & Khasahmadi, 2020), rely on global negative sampling or complex graph augmentation strategies, leading to computational complexity that grows quadratically with the number of nodes, making them difficult to scale to scenarios with millions of nodes. To overcome this limitation, some effort is devoted to negative sampling strategies and model architecture design. For example, cluster-based or subgraph-based local contrastive strategies (Hassani & Khasahmadi, 2020) restrict comparisons to localized neighborhoods to reduce computational costs. Lightweight encoders including parameter-shared GNNs and low-rank approximation designs (Cai et al., 2023) can decrease parameter complexity to some extent.

## 3. CL-GCL

In this section, we first present the preliminaries and notations about GCL. Then, we introduce CL-GCL. Finally, we provide complexity analysis and summarize overall procedure. Figure 1 shows its overview.

### 3.1. Preliminaries

Let $\mathcal{G} = (\boldsymbol{X}, \boldsymbol{A})$ be an undirected graph, where $\boldsymbol{X} \in \mathbb{R}^{N \times D}$ is the embedding matrix, $\boldsymbol{A} \in \{0, 1\}^{N \times N}$ is the adjacency matrix. Let $\mathcal{V}$ and $\mathcal{E}$ denote the node set and the edge set, with the corresponding number $N = |\mathcal{V}|$ and $E = |\mathcal{E}|$, respectively. In general, GCL aims at learning a graph encoder $f_\theta(\boldsymbol{X}, \boldsymbol{A}) \in \mathbb{R}^{N \times D'}$, where $D' \ll D$, to get appropriate node embeddings.

### 3.2. Cross-Community Node Sampling

Given an original graph $\mathcal{G} = (\boldsymbol{X}, \boldsymbol{A})$ and its graph coarsening output $\mathcal{G}^{'} = (\boldsymbol{X}^{'}, \boldsymbol{A}^{'})$ with $N^{'}$ super-nodes and $E^{'}$ super-edges, where $N^{'} \ll N$. The graph coarsening problem requires learning a coarsening matrix $\mathcal{C} \in \{0, 1\}^{N \times N^{'}}$, a linear mapping $\pi : \mathcal{V} \to \mathcal{V}^{'}$. It ensures similar nodes in $\mathcal{G}$ are mapped to the same super-node in $\mathcal{G}^{'}$. Then the coarse graph can be constructed with $\boldsymbol{X}^{'} = \mathcal{C}^{T} \boldsymbol{X}$ and $\boldsymbol{A}^{'} = \mathcal{C}^{T} \boldsymbol{A} \mathcal{C}$. There are several constraints that $\mathcal{C}$ should satisfy, i.e., $\forall i \neq j, \langle \mathcal{C}_i^{T}, \mathcal{C}_j^{T} \rangle = 0$ and $||\mathcal{C}_i^{T}||_0 \geq 1$. The former ensures that each node of $\mathcal{G}$ is mapped to a unique super-node, while the latter requires that each super-node contains at least one node. The goal is to learn matrix $\mathcal{C}$ such that $\mathcal{G}$ and $\mathcal{G}^{'}$ are similar. In this work, we utilize METIS (Karypis & Kumar, 1998) to generate matrix $\mathcal{C}$.

Suppose nodes in the original graph and their community center in the coarse graph are sampled from the same underlying manifold. Assume there are sufficient nodes, i.e., the manifold is well-sampled, we expect each node in the original graph and their community center in the coarse graph to lie on or close to a locally linear patch of the manifold. The optimization objective is as follows

$$\min_{\boldsymbol{w}_i^p} \sum_{i=1}^{K} (||\boldsymbol{x}_i^{'} - \sum_{v_j \in \mathcal{P}_i} w_{ij}^p \boldsymbol{x}_j||_2^2 + \lambda ||\boldsymbol{w}_i^p||_2^2)$$
$$s.t.\ w_{ij}^p \geq 0, i = 1, 2, ..., K, v_j \in \mathcal{P}_i \qquad (1)$$
$$\sum_{v_j \in \mathcal{P}_i} w_{ij}^p = 1, i = 1, 2, ..., K$$

where $\mathcal{P}_i$ is the positive pair set of $v_i$, consisting of all the nodes from the same community of $v_i$ in the original graph. $w_{ij}^p$ is the reconstruction weight to be solved, which can also be regarded as the probability that $v_i$ and $v_j$ are positive pairs. $\lambda$ is a positive hyperparameter to trade off the reconstruction error and regularizer.

Figure 1 gives the equivalent matrix form with $w_{ik}^p = 0, v_k^{'} \in \mathcal{N}_i$. While the optimization problem is a typical quadratic programming problem, there does not exist suitable toolkit on GPU. In the following section, we will give detailed solution.

For a given anchor point $v_i^{'}$, we assign all of the negative pairs the same weight

$$w_{ik}^n = \frac{1}{|\mathcal{N}_i|}, v_k^{'} \in \mathcal{N}_i \qquad (2)$$

where $\mathcal{N}_i$ is the negative pair set of $v_i^{'}$. Similarly, we set $w_{ij}^n = 0, v_j \in \mathcal{P}_i$.

### 3.3. Graph Encoders

We apply GCN (Kipf & Welling, 2017) as graph encoder. It uses several graph convolution layers to aggregate neigh-

borhood information and then updates each layer with the equation as follows

$$\boldsymbol{Z}^{(l+1)} = \sigma \left( \hat{\boldsymbol{A}} \, \boldsymbol{Z}^{(l)} \, \boldsymbol{W}^{(l)} \right) \qquad (3)$$

where $\boldsymbol{Z}^{(l)}$ is the feature matrix at layer $l$, $\boldsymbol{Z}^{(0)} = \boldsymbol{X}$, $\hat{\boldsymbol{A}}$ is the normalized adjacency matrix with self-loops, $\boldsymbol{W}^{(l)}$ denotes the learnable weight matrix of layer $l$ and $\sigma$ is a non-linear activation function, i.e., ReLU.

### 3.4. Community-Node Mixed Contrastive Loss

As Figure 1 illustrates, we construct multiple positive pairs for a given community center coarse graph in the with all the nodes from the same community in the original graph. Then, we use GCN as graph encoder to get the node embeddings $\boldsymbol{Z}, \boldsymbol{Z}^{'}$. In the end, we define the training objective for each anchor point $v_i$ as follows

$$\ell(\boldsymbol{z}_i^{'}) = -\log \frac{\sum\limits_{v_j \in \mathcal{P}_i} w_{ij}^p e^{s(\boldsymbol{z}_i^{'}, \boldsymbol{z}_j)/\tau}}{\sum\limits_{v_j \in \mathcal{P}_i} w_{ij}^p e^{s(\boldsymbol{z}_i^{'}, \boldsymbol{z}_j)/\tau} + \sum\limits_{v_k \in \mathcal{N}_i} w_{ik}^n e^{s(\boldsymbol{z}_i^{'}, \boldsymbol{z}_k^{'})/\tau}}$$
$$(4)$$

where $\tau \in [0, 1]$ is a tunable temperature parameter. The similarity score between node representations is computed as the inner product $s(\boldsymbol{z}_i, \boldsymbol{z}_j) = \boldsymbol{z}_i^{\top} \boldsymbol{z}_j$.

Finally, we define the overall loss as follows

$$\mathcal{L} = \frac{1}{K} \sum_{i=1}^{K} \ell(\boldsymbol{z}_i^{'}) \qquad (5)$$

### 3.5. Theoretical Analysis

In this section, we first prove that the contrastive loss on the original graph is close to the sum of the community-node mixed contrastive loss and the low-rank approximation gap. In other words, if the low-rank approximation adequately satisfies, we can estimate the original GCL loss with the anchor-based contrastive loss. Then, we give the closed form solution of $\boldsymbol{w}^p$ without constraints. On this basis, we derive the optimal solution with constraints.

**Definition 1.** We define *node-level contrastive loss* as

$$\mathcal{L}_{node}(\boldsymbol{W})$$
$$= \mathbb{E}_{(v_i, v_j) \in \mathcal{P}_{\text{pos}}} \left[ -\log \frac{e^{s(\boldsymbol{z}_i, \boldsymbol{z}_j)/\tau}}{e^{s(\boldsymbol{z}_i, \boldsymbol{z}_j)/\tau} + \sum\limits_{v_k \in \mathcal{N}_i} e^{s(\boldsymbol{z}_i, \boldsymbol{z}_k)/\tau}} \right]$$
$$= \sum_{i,j} a_{ij} \cdot \ell(\boldsymbol{z}_i, \boldsymbol{z}_j)$$

where $\boldsymbol{z}_i = f(\boldsymbol{x}_i; \boldsymbol{W})$ denotes the representation of node $v_i$ computed by an encoder (e.g., GCN) given the input

features $x_i$. The set of positive node pairs (typically edges in the graph) is denoted as $\mathcal{P}_{\text{pos}}$, and $\mathcal{N}_i$ represents the set of negative pairs for node $v_i$. The temperature parameter $\tau$ is used to scale the similarity scores. $a_{ij}$ denotes the adjacency matrix element indicating whether nodes $v_i$ and $v_j$ form a positive pair ($a_{ij} = 1$ if connected).

**Definition 2.** We define *community-node mixed contrastive loss* as

$$\mathcal{L}_{com-node}(\boldsymbol{W})$$

$$= \mathbb{E}_{\mathcal{P}_i \in \mathcal{K}} \left[ -\log \frac{\sum\limits_{v_j \in \mathcal{P}_i} w^p_{ij} \cdot e^{s(\boldsymbol{z}'_i, \boldsymbol{z}_j)/\tau}}{\sum\limits_{v_j \in \mathcal{P}_i} w^p_{ij} \cdot e^{s(\boldsymbol{z}'_i, \boldsymbol{z}_j)/\tau} + \sum\limits_{v_k \in \mathcal{N}_i} w^n_{ik} \cdot e^{s(\boldsymbol{z}'_i, \boldsymbol{z}'_k)/\tau}} \right]$$

$$= \frac{1}{|\mathcal{K}|} \sum_{\mathcal{P}_i \in \mathcal{K}} \left[ \sum_{v_j \in \mathcal{P}_i} w^p_{ij} \cdot \ell(\boldsymbol{z}'_i, \boldsymbol{z}_j) + \sum_{v_k \in \mathcal{N}_i} w^n_{ik} \cdot \ell(\boldsymbol{z}'_i, \boldsymbol{z}'_k) \right],$$

where $\boldsymbol{z}'_i = \frac{1}{|\mathcal{P}_i|} \sum\limits_{v_j \in \mathcal{P}_i} \boldsymbol{z}_j$ denotes the mean feature (anchor) of community $\mathcal{P}_i$, and let $\boldsymbol{z}_j = f(\boldsymbol{x}_j; \boldsymbol{W})$ be the encoded representation of node $v_j$. The term $w^p_{ij}$ is the normalized weight assigned to node $v_j$ within its own community $\mathcal{P}_i$, satisfying $\sum\limits_{v_j \in \mathcal{P}_i} w^p_{ij} = 1$. Similarly, $w^n_{ik}$ is the normalized weight assigned to a negative community $v_k \in \mathcal{N}_i$, such that $\sum\limits_{v_k \in \mathcal{N}_i} w^n_{ik} = 1$. The temperature hyperparameter is denoted by $\tau$, which controls the sharpness of the similarity distribution.

**Definition 3.** We define *community-level contrastive loss* as

$$\mathcal{L}_{com}(\boldsymbol{W}) = \mathbb{E}_{\mathcal{P}_i \in \mathcal{K}} \left[ -\log \frac{e^{s(\boldsymbol{z}'_i, \boldsymbol{z}'_i)/\tau}}{e^{s(\boldsymbol{z}'_i, \boldsymbol{z}'_i)/\tau} + \sum\limits_{v_k \in \mathcal{N}_i} e^{s(\boldsymbol{z}'_i, \boldsymbol{z}'_k)/\tau}} \right]$$

$$= \sum_{i,j} (\mathcal{C}\mathcal{C}^\top)_{ij} \cdot \ell(\boldsymbol{z}_i, \boldsymbol{z}_j)$$

$$= \frac{1}{|\mathcal{K}|} \sum_{\mathcal{P}_i \in \mathcal{K}} \left[ \ell(\boldsymbol{z}'_i, \boldsymbol{z}'_i) + \sum_{v_k \in \mathcal{N}_i} \ell(\boldsymbol{z}'_i, \boldsymbol{z}'_k) \right],$$

where $\boldsymbol{z}'_i \in \mathbb{R}^{D'}$ denotes the mean feature vector of community $\mathcal{P}_i$, and let $\boldsymbol{z}'_i = f(\boldsymbol{x}'_i; \boldsymbol{W})$ be the encoded representation of the community anchor using an encoder $f$ with parameters $\boldsymbol{W}$. The temperature hyperparameter $\tau$ is used to scale the similarity scores. Let $\mathcal{P}$ be the set of all communities (partitions). In this idealized setting, positive pairs are self-pairs $(\boldsymbol{z}'_i, \boldsymbol{z}'_i)$, while negative pairs are inter-community pairs $(\boldsymbol{z}'_i, \boldsymbol{z}'_k)$ with $k \neq i$. The term $(\mathcal{C}\mathcal{C}^\top)_{ij}$ indicates the affinity or similarity between community $i$ and community $j$. This loss assumes an ideal case where all nodes in a community share the same feature, and thus the community anchor fully represents the community.

**Proposition 1.** *Let $\mathcal{G} = (\boldsymbol{X}, \boldsymbol{A})$ be a graph generated by the Stochastic Block Model (SBM), where the graph is parti-*

tioned into $K$ disjoint communities $\mathcal{K} = \{\mathcal{P}_1, \mathcal{P}_2, \ldots, \mathcal{P}_K\}$, where each community $\mathcal{P}_i \subseteq \mathcal{V}$. For any node pair $(v_i, v_j) \in \mathcal{P}_i$, an edge exists between them with probability $p_{in}$, modeling the intra-community connectivity. For any node pair $v_i \in \mathcal{P}_k, v_j \in \mathcal{P}_l$ with $k \neq l$, an edge exists between them with probability $p_{out}$, capturing the inter-community connectivity. The model assumes a strong community structure, meaning that the edge probabilities satisfy $p_{in} \gg p_{out}$, indicating that intra-community edges are significantly denser than inter-community edges.

Here we consider the *SBM* graph, denoted as $\mathcal{G} = (\boldsymbol{X}, \boldsymbol{A})$. We use $\|\cdot\|_2$ to represent $l_2$ norm and $\|\cdot\|_F$ to represent Frobenius norm. Additionally, we denote the feature vector of each node as $\boldsymbol{x}_i \in \mathbb{R}^D$. Then we can prove the following theorem. We leave the proof and details of notations in Appendix C.

**Theorem 1.** *For the SBM graph $\mathcal{G} = (\boldsymbol{X}, \boldsymbol{A})$, we construct an even partition $\mathcal{K} = \{\mathcal{P}_1, \mathcal{P}_2, \ldots, \mathcal{P}_K\}$, with the corresponding community assignment matrix $\mathcal{C} \in \{0,1\}^{N \times N'}$. The partition residual matrix is defined as $\boldsymbol{R} = \boldsymbol{A} - \mathcal{C}\mathcal{C}^\top$. Let the encoder $f(\cdot; \boldsymbol{W})$ be a GCN with $l$ layers, and let the total weight matrix be $\boldsymbol{W}_{total} = \boldsymbol{W}^{(l)} \boldsymbol{W}^{(l-1)} \cdots \boldsymbol{W}^{(1)}$. Assume that node features within each community $\mathcal{P}_i$ satisfy a local smoothness condition such that $\max\limits_{v_j \in \mathcal{P}_i} \|\boldsymbol{x}_j - \boldsymbol{z}'_i\|_2 \leq \epsilon, \boldsymbol{z}'_i = f(\frac{1}{|\mathcal{P}_i|} \sum\limits_{v_j \in \mathcal{P}_i} \boldsymbol{x}_j; \boldsymbol{W})$. where $\boldsymbol{z}'_i$ denotes the mean feature vector of nodes in community $\mathcal{P}_i$. Then, the difference between the original node-level contrastive loss $\mathcal{L}_{node}(\boldsymbol{W})$ and the community anchor-level contrastive loss $\mathcal{L}_{com-node}(\boldsymbol{W})$ is bounded by*

$$|\mathcal{L}_{node}(\boldsymbol{W}) - \mathcal{L}_{com-node}(\boldsymbol{W})|$$
$$\leq C_1 \cdot \|\boldsymbol{R}\|_F \cdot S_X \cdot \|\boldsymbol{W}_{total}\|_2 + C_2 \cdot \epsilon \cdot \|\boldsymbol{W}_{total}\|_2$$

*where $S_X = \max_j \|\boldsymbol{x}_j\|_2$ is the maximum norm of node features; $C_1, C_2$ are constants related to the graph size and the temperature parameter $\tau$.*

**Proposition 2** ((Liang et al., 2025a)). *The optimal closed solution of Equation 1 without constraints is $\boldsymbol{w}^p_i = (\boldsymbol{X}_i \boldsymbol{X}^T_i + \lambda \boldsymbol{I})^{-1} \boldsymbol{X}_i \boldsymbol{x}'_i$, where $\boldsymbol{w}^p_i \in \mathbb{R}^{|\mathcal{P}_i|}$ is the reconstruction weight vector of $v_i$, $\boldsymbol{X}_i \in \mathbb{R}^{|\mathcal{P}_i| \times D}$ is the positive pairs embedding matrix of $v_i$ in the original graph and $\boldsymbol{x}'_i \in \mathbb{R}^D$ is the embedding vector of $v_i$ in the coarse graph. With non-negativity and normalization constraints, we can derive its optimal closed solution via the ReLU activation function, i.e., $\boldsymbol{w}^p_i = \text{ReLU}(\boldsymbol{w}^p_i)$ and normalization, i.e.,*

$$\boldsymbol{w}^p_i = \boldsymbol{w}^p_i / \sum_{j=1}^N w^p_{ij}.$$

### 3.6. Complexity Analysis

In CL-GCL, given a graph with $N$ nodes, $M$ edges and its coarsened graph with $N'$ nodes, $M'$ edges. Suppose there

*Table 1.* Node classification accuracy (mean ± std) (%) on 6 homophilic datasets. Best results are colored: **first**, **second**, **third**.

| Model | Cora | Citeseer | Pubmed | Amazon-Photo | Amazon-Computers | Wiki-CS |
|---|---|---|---|---|---|---|
| GCN (Kipf & Welling, 2017) | 79.6 ± 1.8 | 66.0 ± 1.2 | 79.0 ± 2.5 | 86.3 ± 1.6 | 76.4 ± 1.8 | 67.3 ± 1.5 |
| GAT (Velickovic et al., 2018) | 81.2 ± 1.6 | 68.9 ± 1.8 | 78.5 ± 1.8 | 86.5 ± 2.1 | 77.9 ± 1.8 | 68.6 ± 1.9 |
| CGPN (Wan et al., 2021b) | 74.0 ± 1.7 | 63.7 ± 1.6 | 73.3 ± 2.5 | 84.1 ± 1.5 | 74.7 ± 1.3 | 66.1 ± 2.1 |
| CG3 (Wan et al., 2021a) | 80.6 ± 1.6 | 70.9 ± 1.5 | 78.9 ± 2.6 | 89.4 ± 1.9 | 77.8 ± 1.7 | 68.0 ± 1.5 |
| DGI (Velickovic et al., 2019) | 82.1 ± 1.3 | 71.6 ± 1.2 | 78.3 ± 2.4 | 83.5 ± 1.2 | 78.8 ± 1.1 | 69.1 ± 1.4 |
| GMI (Peng et al., 2020) | 79.4 ± 1.2 | 66.9 ± 2.2 | 76.8 ± 2.3 | 86.7 ± 1.5 | 76.1 ± 1.2 | 67.8 ± 1.8 |
| MVGRL (Hassani & Khasahmadi, 2020) | 82.4 ± 1.5 | 71.1 ± 1.4 | 79.5 ± 2.2 | 89.7 ± 1.2 | 78.7 ± 1.7 | 69.2 ± 1.2 |
| GRACE (Zhu et al., 2020) | 79.6 ± 1.4 | 67.0 ± 1.7 | 74.6 ± 3.5 | 87.9 ± 1.4 | 76.8 ± 1.7 | 67.8 ± 1.4 |
| GCA (Zhu et al., 2021) | 79.0 ± 1.4 | 65.6 ± 2.4 | 81.5 ± 2.5 | 87.0 ± 1.9 | 76.9 ± 1.4 | 67.6 ± 1.3 |
| SUGRL (Mo et al., 2022) | 81.3 ± 1.2 | 71.0 ± 1.8 | 80.5 ± 1.6 | 90.5 ± 1.9 | 78.2 ± 1.2 | 68.7 ± 1.1 |
| AFGRL (Lee et al., 2022) | 78.6 ± 1.3 | 70.8 ± 2.1 | 76.4 ± 2.5 | 89.2 ± 1.1 | 77.7 ± 1.1 | 68.0 ± 1.7 |
| NCLA (Shen et al., 2023) | 82.2 ± 1.6 | 71.7 ± 0.9 | 82.0 ± 1.4 | 90.2 ± 1.3 | 79.8 ± 1.5 | 70.3 ± 1.7 |
| GTCA (Liang et al., 2025b) | 82.5 ± 1.3 | 69.7 ± 1.7 | 79.8 ± 1.3 | 90.5 ± 1.2 | 79.2 ± 1.4 | 69.7 ± 1.5 |
| E2Neg (Huang et al., 2025) | 81.2 ± 1.4 | 71.0 ± 1.4 | 79.9 ± 1.7 | 90.2 ± 1.5 | 80.1 ± 1.2 | 70.5 ± 1.6 |
| EPAGCL (Xu et al., 2025) | 82.5 ± 1.3 | 71.1 ± 1.5 | 81.5 ± 1.1 | 89.9 ± 1.2 | 80.2 ± 1.4 | 70.3 ± 1.5 |
| CL-GCL$_{\text{METIS}}$ | 82.7 ± 1.3 | 72.6 ± 1.6 | 81.8 ± 1.7 | 90.5 ± 0.9 | 80.5 ± 1.2 | 70.9 ± 1.9 |

is a simple encoder which computes embeddings in time and space $O(M + N)$ and $O(M' + N')$, respectively. This property is satisfied by most popular GNN architectures, such as GCN (Kipf & Welling, 2017) and GAT (Velickovic et al., 2018). Before training, we obtain the reconstruction weights between each anchor and all nodes within its corresponding community, which can be ignored in the subsequent complexity analysis. In each update step, CL-GCL performs 2 encoder computations (once for each graph) and backpropagates the learning signal twice (once for each graph), plus a prediction step. We assume the backward pass to be approximately as costly as the forward pass and ignore the cost of graph partition in this analysis. Therefore, the total time and space complexity per update step for CL-GCL is $2C_{\text{encoder}}(M + N) + 2C_{\text{encoder}}(M' + N') + 2C_{\text{prediction}}N + 2C_{\text{prediction}}N' + C(N + N'^2)$, where $C$ is a constant dependent on architecture of different components. The terms correspond to the computational complexity of the original and coarsened graphs in GNN encoding phase and prediction head, as well as loss function computation, respectively.

We summarize the overall procedure of CL-GCL in Appendix A.

# 4. Experiments

In this section, we evaluate both the effectiveness and efficiency of CL-GCL$_{\text{METIS}}$ on 14 benchmark datasets of different sizes. We implement all experiments on the platform with PyTorch 2.0.1 and PyTorch Geometric 2.6.1 on NVIDIA 3090 GPU (24GB memory) and an Intel(R) Xeon(R) Gold 6254 CPU @ 3.10GHz. To ensure repro-

ducibility, we summarize detailed hyperparameters specifications in Appendix D. More empirical results and analysis are included in Appendix E.

## 4.1. Datasets

We conduct experiments on 14 widely used benchmark datasets. The homophilic datasets include 6 small-scale and medium-scale datasets, i.e., Cora, Citeseer, Pubmed, Amazon-Photo, Amazon-Computers, Wiki-CS as well as 3 large-scale dataset Ogbn-Arxiv, Ogbn-Products and Ogbn-Papers100M. The heterophilic graph datasets include Squirrel, Actor, Cornell, Texas, and Wisconsin. The statistics are summarized in Appendix B. For node classification task, we split Cora, Citeseer and Pubmed following (Yang et al., 2016), and Amazon-Photo, Amazon-Computers, Wiki-CS following (Liu et al., 2020). We follow the data splits of GRASS (Yang et al., 2025) on heterophilic graph datasets. For link prediction task, we follow the experimental setup of GCA (Zhu et al., 2021). For Ogbn-Arxiv, Ogbn-Products and Ogbn-Papers100M, we follow the public split of OGB (Hu et al., 2020).

## 4.2. Node Classification

Baselines on 6 homophilic graph datasets include 2 semi-supervised GNNs, i.e., GCN (Kipf & Welling, 2017), GAT (Velickovic et al., 2018), 2 semi-supervised GCL methods, i.e., CGPN (Wan et al., 2021b), CG3 (Wan et al., 2021a), and 9 self-supervised GCL methods, i.e., DGI (Velickovic et al., 2019), GMI (Peng et al., 2020), MVGRL (Hassani & Khasahmadi, 2020), GRACE (Zhu et al., 2020), GCA (Zhu et al., 2021), SUGRL (Mo et al., 2022), AFGRL (Lee et al., 2022), NCLA (Shen et al., 2023), GTCA (Liang et al.,

*Table 2.* Node classification accuracy (mean ± std) (%) on 5 heterophilic graph datasets. Best results are colored: **first**, **second**, **third**.

| Method | Squirrel | Actor | Cornell | Texas | Wisconsin |
|---|---|---|---|---|---|
| GCN (Kipf & Welling, 2017) | 36.3 ± 1.5 | 30.8 ± 0.8 | 57.0 ± 3.3 | 60.0 ± 4.8 | 56.5 ± 6.6 |
| GAT (Velickovic et al., 2018) | 32.1 ± 3.3 | 28.1 ± 1.5 | 59.5 ± 3.6 | 61.6 ± 3.8 | 54.7 ± 6.9 |
| DeepWalk (Perozzi et al., 2014) | 32.9 ± 1.6 | 22.8 ± 0.6 | 63.4 ± 4.6 | 60.6 ± 7.6 | 55.4 ± 6.0 |
| Node2Vec (Grover & Leskovec, 2016) | 22.8 ± 0.7 | 28.3 ± 1.3 | 42.9 ± 7.5 | 41.9 ± 7.8 | 37.5 ± 7.1 |
| DGI (Velickovic et al., 2019) | 31.8 ± 0.8 | 29.8 ± 0.7 | 63.4 ± 4.6 | 60.6 ± 7.6 | 55.4 ± 6.0 |
| GMI (Peng et al., 2020) | 30.1 ± 1.9 | 27.8 ± 0.9 | 54.8 ± 5.1 | 50.5 ± 2.2 | 46.0 ± 2.8 |
| GRACE (Zhu et al., 2020) | 31.3 ± 1.2 | 29.0 ± 0.8 | 54.9 ± 7.0 | 57.6 ± 5.7 | 50.0 ± 5.8 |
| MVGRL (Hassani & Khasahmadi, 2020) | 35.5 ± 1.3 | 30.0 ± 0.7 | 64.3 ± 5.4 | 62.4 ± 5.6 | 62.4 ± 4.3 |
| GCA (Zhu et al., 2021) | 35.5 ± 0.9 | 29.7 ± 1.5 | 55.4 ± 4.6 | 59.5 ± 6.2 | 50.8 ± 4.1 |
| BGRL (Thakoor et al., 2022) | 32.6 ± 0.8 | 29.9 ± 0.8 | 57.3 ± 5.5 | 59.2 ± 5.9 | 52.4 ± 4.1 |
| AF-GCL (Wang et al., 2022a) | 40.6 ± 0.7 | 32.3 ± 0.7 | 64.8 ± 3.6 | 68.9 ± 2.2 | 67.8 ± 3.3 |
| HLCL (Yang & Mirzasoleiman, 2024) | 34.1 ± 0.9 | 31.0 ± 0.9 | 58.5 ± 3.2 | 65.3 ± 1.8 | 67.0 ± 3.8 |
| MaskGAE (Li et al., 2023a) | 37.9 ± 1.2 | 31.1 ± 0.9 | 55.4 ± 4.6 | 66.0 ± 6.5 | 56.9 ± 4.5 |
| GraphACL (Xiao et al., 2023) | 41.4 ± 4.6 | 30.1 ± 0.2 | 59.7 ± 1.4 | 70.2 ± 1.6 | 68.3 ± 2.0 |
| SGCL (Sun et al., 2024) | 38.5 ± 0.9 | 31.2 ± 0.6 | 62.8 ± 2.9 | 68.3 ± 4.3 | 63.3 ± 3.1 |
| GCIL (Mo et al., 2024) | 40.7 ± 1.4 | 31.7 ± 0.9 | 63.6 ± 3.2 | 67.5 ± 6.3 | 65.2 ± 1.9 |
| GRASS (Yang et al., 2025) | 40.4 ± 2.6 | 34.9 ± 0.9 | 70.3 ± 5.7 | 77.4 ± 2.2 | 77.1 ± 6.0 |
| CL-GCL$_{\text{METIS}}$ | 42.0 ± 3.1 | 35.5 ± 1.5 | 70.4 ± 6.1 | 71.1 ± 6.4 | 68.6 ± 4.2 |

*Table 3.* Link prediction results (%) on 6 homophilic datasets. Best results are colored: **first**, **second**, **third**.

| Model | Cora | | Citeseer | | Pubmed | | Amazon-Photo | | Amazon-Computers | | Wiki-CS | |
|---|---|---|---|---|---|---|---|---|---|---|---|---|
| | AUC | AP | AUC | AP | AUC | AP | AUC | AP | AUC | AP | AUC | AP |
| Spectral (Ng et al., 2001) | 84.6 | 88.5 | 80.5 | 85 | 84.2 | 87.8 | 83.6 | 84.3 | 86.7 | 87.3 | 81.2 | 80.3 |
| DeepWalk (Perozzi et al., 2014) | 83.1 | 85 | 80.5 | 83.6 | 84.4 | 84.1 | 83.2 | 85.1 | 87.2 | 87.5 | 82.1 | 81.5 |
| GAE (Schulman et al., 2016) | 91 | 92 | 89.5 | 89.9 | 96.4 | 96.5 | 89.9 | 89.6 | 93 | 93.2 | 83.2 | 82.1 |
| VGAE (Kipf & Welling, 2016) | 91.4 | 92.6 | 90.8 | 92 | 94.4 | 94.7 | 89.3 | 88.8 | 92.6 | 92.8 | 82.5 | 83.4 |
| ARGE (Pan et al., 2018) | 92.4 | 93.2 | 91.9 | 93 | 96.8 | 97.1 | 91.5 | 92.2 | 92.5 | 92.7 | 83.5 | 84.2 |
| ARVGA (Pan et al., 2018) | 92.4 | 92.6 | 92.4 | 93 | 96.5 | 96.8 | 92.1 | 91.9 | 93.1 | 92.8 | 82.6 | 83.9 |
| GRACE (Zhu et al., 2020) | 90.9 | 91 | 92.1 | 92.2 | 97 | 97.1 | 90.8 | 89.3 | 91.6 | 91.2 | 89.5 | 88.5 |
| GCA (Zhu et al., 2021) | 91.4 | 91.5 | 92 | 92.6 | 96.3 | 96.5 | 92.3 | 91.3 | 92.5 | 91.5 | 86.4 | 86.3 |
| GDCL (Zhao et al., 2021) | 91.7 | 90.9 | 91.9 | 92 | 96.5 | 96.3 | 93.1 | 93.6 | 93.1 | 93.2 | 85.2 | 84.6 |
| ProGCL (Xia et al., 2022b) | 92.9 | 93.5 | 93.1 | 93.3 | 96.1 | 96.7 | 92.6 | 93.5 | 94.5 | 94.2 | 83.6 | 84.1 |
| AUGCL (Niu et al., 2024) | 93.3 | 93.2 | 92.5 | 92.8 | 96.3 | 96.5 | 94.2 | 93.9 | 93.7 | 93.9 | 88.9 | 88.5 |
| E2Neg (Huang et al., 2025) | 97.1 | 94.7 | 98.3 | 97.2 | 96.9 | 93.1 | 95.9 | 91.8 | 94.8 | 90.3 | 93.7 | 88.6 |
| EPAGCL (Xu et al., 2025) | 97.7 | 95.6 | 98.3 | 97.1 | 96.8 | 92.3 | 92.8 | 86.3 | 92.6 | 87.0 | 93.6 | 87.7 |
| CL-GCL$_{\text{METIS}}$ | 97.6 | 95.1 | 98.2 | 95.8 | 98.6 | 97.0 | 94.6 | 93.7 | 94.5 | 93.9 | 93.8 | 88.8 |

2025b), E2Neg (Huang et al., 2025) and EPAGCL (Xu et al., 2025).

Baselines on 5 heterophilic graph datasets include 2 semi-supervised GNN models GCN (Kipf & Welling, 2017) and GAT (Velickovic et al., 2018), unsupervised graph learning methods DeepWalk (Perozzi et al., 2014) and Node2Vec (Grover & Leskovec, 2016), self-supervised graph learning methods DGI (Velickovic et al., 2019), GMI (Peng et al., 2020), GRACE (Zhu et al., 2020), MVGRL (Hassani & Khasahmadi, 2020), GCA (Zhu et al., 2021), BGRL (Thakoor et al., 2022), AF-GCL (Wang et al., 2022a), HLCL (Yang & Mirzasoleiman, 2024), MaskGAE (Li et al., 2023a), GraphACL (Xiao et al., 2023), SGCL (Sun et al., 2024),

GCIL (Mo et al., 2024) and GRASS (Yang et al., 2025).

Tables 1-2 lists the node classification accuracy of CL-GCL$_{\text{METIS}}$ and baselines with a logistic regression model on commonly used homophilic and heterophilic graph datasets, respectively. We observe that CL-GCL$_{\text{METIS}}$ achieves state-of-the-art results with respect to existing GCL methods. On the whole, CL-GCL$_{\text{METIS}}$ ranks first on 5 homophilic, 3 heterophilic datasets and second on 1 homophilic, 2 heterophilic datasets.

*Table 4.* Ablation study on node classification task for positive/negative pairs weights. The metric is the Accuracy (mean ± std) (%).

| $W^p$ | $W^n$ | Cora | Citeseer | Pubmed | Amazon-Photo | Amazon-Computers | Wiki-CS |
|---|---|---|---|---|---|---|---|
| − | − | $81.5 \pm 1.1$ | $69.7 \pm 1.7$ | $79.0 \pm 2.1$ | $89.4 \pm 1.1$ | $79.0 \pm 1.8$ | $69.1 \pm 1.9$ |
| − | ✓ | $81.8 \pm 1.9$ | $70.6 \pm 1.4$ | $79.7 \pm 1.4$ | $89.7 \pm 1.0$ | $79.4 \pm 1.7$ | $69.6 \pm 1.8$ |
| ✓ | − | $82.2 \pm 1.4$ | $72.0 \pm 1.6$ | $79.9 \pm 2.3$ | $89.9 \pm 0.7$ | $80.1 \pm 1.1$ | $70.5 \pm 1.8$ |
| ✓ | ✓ | $\mathbf{82.7 \pm 1.3}$ | $\mathbf{72.6 \pm 1.6}$ | $\mathbf{81.8 \pm 1.7}$ | $\mathbf{90.5 \pm 0.9}$ | $\mathbf{80.5 \pm 1.2}$ | $\mathbf{70.9 \pm 1.9}$ |

*Table 5.* Time (s/epoch) usage across 6 homophilic datasets on node classification task. Best results are colored: first, second, third. Improve means how many times is CL-GCL$_{\text{METIS}}$ faster than baselines. '-' means the improvement range.

| Model | Cora | Citeseer | Pubmed | Amazon-Photo | Amazon-Computers | Wiki-CS |
|---|---|---|---|---|---|---|
| GRACE (Zhu et al., 2020) | 0.017 | 0.022 | 0.271 | 0.045 | 0.126 | 0.104 |
| GCA (Zhu et al., 2021) | 0.014 | 0.018 | 0.232 | 0.062 | 0.145 | 0.118 |
| AFGRL (Lee et al., 2022) | 0.584 | 0.766 | 9.281 | 2.214 | 4.319 | 3.414 |
| NCLA (Shen et al., 2023) | 0.018 | 0.027 | 0.488 | 0.084 | 0.232 | 0.239 |
| GTCA (Liang et al., 2025b) | 2.512 | 3.974 | 14.132 | 6.935 | 10.903 | 8.261 |
| E2Neg (Huang et al., 2025) | 0.012 | 0.013 | 0.026 | 0.034 | 0.037 | 0.055 |
| EPAGCL (Xu et al., 2025) | 0.081 | 0.117 | 0.512 | 0.201 | 0.341 | 0.301 |
| CL-GCL$_{\text{METIS}}$ | 0.009 | 0.010 | 0.023 | 0.031 | 0.035 | 0.050 |
| Improve | 1.3-279.1× | 1.3-397.4× | 1.1-614.5× | 1.1-224.4× | 1.1-312.4× | 1.1-165.6× |

*Table 6.* The GPU memory (GiBs) usage across 6 homophilic datasets on node classification task. Best results are colored: first, second, third. Improve means how lightweight is CL-GCL$_{\text{METIS}}$ compared with other baselines. '-' means the improvement range.

| Model | Cora | Citeseer | Pubmed | Amazon-Photo | Amazon-Computers | Wiki-CS |
|---|---|---|---|---|---|---|
| GRACE (Zhu et al., 2020) | 0.47 | 0.55 | 13.17 | 2.11 | 6.23 | 4.46 |
| GCA (Zhu et al., 2021) | 0.58 | 0.69 | 14.32 | 2.34 | 6.61 | 5.01 |
| AFGRL (Lee et al., 2022) | 0.37 | 0.48 | 12.59 | 2.12 | 5.94 | 4.41 |
| NCLA (Shen et al., 2023) | 0.67 | 1.02 | 14.56 | 2.25 | 7.16 | 7.71 |
| GTCA (Liang et al., 2025b) | 1.64 | 2.64 | 22.03 | 4.09 | 13.14 | 10.89 |
| E2Neg (Huang et al., 2025) | 0.10 | 0.17 | 0.44 | 0.21 | 0.30 | 0.24 |
| EPAGCL (Xu et al., 2025) | 0.12 | 0.20 | 0.52 | 0.23 | 0.32 | 0.27 |
| CL-GCL$_{\text{METIS}}$ | 0.08 | 0.16 | 0.42 | 0.18 | 0.28 | 0.23 |
| Improve | 20.0-95.1% | 5.9-93.9% | 4.5-98.1% | 14.3-95.6% | 6.7-97.9% | 4.2-97.9% |

### 4.3. Link Prediction

Baselines on link prediction task include Spectral (Ng et al., 2001), DeepWalk (Perozzi et al., 2014), GAE (Schulman et al., 2016), VGAE (Kipf & Welling, 2016), ARGE (Pan et al., 2018), ARVGA (Pan et al., 2018), GRACE (Zhu et al., 2020), GCA (Zhu et al., 2021), GDCL (Zhao et al., 2021), ProGCL (Xia et al., 2022b), AUGCL (Niu et al., 2024), E2Neg (Huang et al., 2025) and EPAGCL (Xu et al., 2025). We use the Area Under Curve (AUC) and Average Precision (AP) to evaluate the performance.

Table 3 shows that CL-GCL$_{\text{METIS}}$ consistently outperforms all the other baselines under different evaluation criteria, highlighting its potential on link prediction task. For in-

stance, it achieves 98.6% AUC on Pubmed, i.e., a 1.6% relative improvement over existing state-of-the-art.

### 4.4. Ablation Study

To study the impact of each component, we remove positive/negative pairs weights of CL-GCL$_{\text{METIS}}$. The removal of either positive pairs weights $W^p$ or negative pairs weights $W^n$ leads to poorer performance as shown in Table 4, which demonstrates the rationality and validity of CL-GCL$_{\text{METIS}}$. In addition, it is noteworthy that compared with the negative pairs weight matrix $W^n$, the positive pairs weight matrix $W^p$ is more crucial for CL-GCL$_{\text{METIS}}$, mainly due to its great significance in global nonlinear structure recovery.

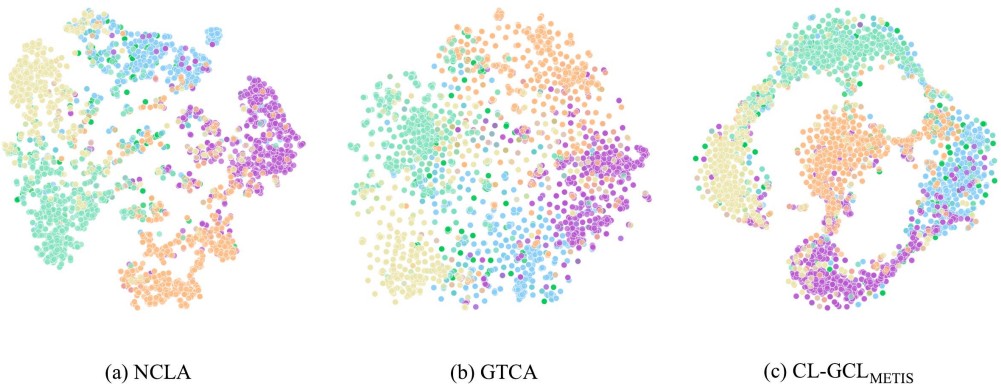

(a) NCLA      (b) GTCA      (c) CL-GCL$_{\text{METIS}}$

*Figure 2.* Visualization of NCLA, GTCA and CL-GCL$_{\text{METIS}}$ embeddings on Citeseer dataset.

### 4.5. Time and Memory Usage

As Tables 5-6 show, CL-GCL$_{\text{METIS}}$ is consistently more efficient than other self-supervised baselines in both time and memory usage. Remarkably, CL-GCL$_{\text{METIS}}$ is 614.5 times faster on Pubmed for time per epoch at most, and costs 20.0% less memory on Cora for GPU memory usage than the most lightweight baseline E2Neg (Huang et al., 2025). Such amazing boost of time and memory efficiency can be contributed to graph coarsening and distance calculation removal, which significantly reduces the number of positive/negative pairs and avoids large-scale pairwise similarity computations.

### 4.6. Comparison with Graph Coarsening Methods

We conduct additional experiments to investigate the impact of graph coarsening on the performance of CL-GCL. In Table 7, we explore the effects of different graph coarsening methods including variation neighborhoods (Loukas, 2019), algebraic JC (Ron et al., 2011) and affinity GS (Livne & Brandt, 2012), on the node classification performance of CL-GCL. Experimental results reveal that graph coarsening methods have little impact on CL-GCL.

*Table 7.* Node classification accuracy (mean ± std) (%) of different graph coarsening methods.

| Method | Cora | Citeseer | Pubmed |
|---|---|---|---|
| CL-GCL$_{\text{VN}}$ | 82.6 ± 1.4 | 72.5 ± 1.3 | 81.0 ± 1.8 |
| CL-GCL$_{\text{JC}}$ | 82.7 ± 1.5 | 72.3 ± 1.3 | 81.5 ± 1.7 |
| CL-GCL$_{\text{GS}}$ | 82.1 ± 1.6 | 72.1 ± 1.0 | 81.2 ± 1.5 |
| CL-GCL$_{\text{METIS}}$ | 82.7 ± 1.3 | 72.6 ± 1.6 | 81.8 ± 1.7 |

### 4.7. Visualization

We utilize t-SNE (Van der Maaten & Hinton, 2008) to provide a more intuitive node embeddings visualization of NCLA (Shen et al., 2023), GTCA (Liang et al., 2025b)

and CL-GCL$_{\text{METIS}}$ on Citeseer dataset as shown in Figure 2. Different colors represent different classes. It is clear that compared with GCL baselines, CL-GCL$_{\text{METIS}}$ is much more effective in distinguishing different classes.

## 5. Conclusion

In this paper, we introduce CL-GCL, a novel, comprehensive and lightweight GCL framework to learn effective node representations from graphs. CL-GCL designs a cross-community node sampling strategy, achieving global non-linear structure recovery and computational efficiency optimization. Theoretical analysis prove that CL-GCL implicitly optimizes the original contrastive loss more efficiently. Extensive experiments on real-world small and large-scale graphs demonstrate its advantages over current methods.

In future, we will extend such unsupervised contrastive paradigms to lightweight incremental spatial-temporal modeling and multimodal data for efficient computation in large-scale graph scenarios.

## Acknowledgements

This work is supported by National Natural Science Foundation of China (No.62376142, U25A20529).

## Impact Statement

This paper presents work whose goal is to advance the field of Machine Learning. There are many potential societal consequences of our work, none which we feel must be specifically highlighted here.

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

## A. Algorithm

---

**Algorithm 1** CL-GCL

---

**Input**: Graph $\mathcal{G} = (\mathcal{V}, \mathcal{E})$
**Output**: Node embeddings $\boldsymbol{Z}, \boldsymbol{Z}'$

 1: Conduct graph coarsening to obtain $\mathcal{G}' = (\boldsymbol{X}', \boldsymbol{A}')$;
 2: **for** i in 1 to $K$ **do**
 3:     Compute the community center $\boldsymbol{z}_i'$ in the coarse graph;
 4:     Construct positive pair set $\mathcal{P}_i$ of $\boldsymbol{z}_i'$ with nodes from the same community in the original graph;
 5:     Construct negative pair set $\mathcal{N}_i$ of $\boldsymbol{z}_i'$ with other community centers in the coarse graph;
 6: **end for**
 7: Compute positive pairs weight matrix $\boldsymbol{W}^p$ as in Proposition 2;
 8: Compute negative pairs weight matrix $\boldsymbol{W}^n$ as in Equation 2;
 9: Initialize GNN encoder paramenters $\{\boldsymbol{W}, \boldsymbol{b}\}$;
10: **while** not converge **do**
11:     Obtain GNN embeddings $\boldsymbol{Z} = f_\theta(\mathcal{G})$ and $\boldsymbol{Z}' = f_\theta(\mathcal{G}')$;
12:     Do forward pass, compute $\mathcal{L}$ as in Equations 4-5;
13:     Do backward propagation with $\mathcal{L}$;
14: **end while**
15: **return** $\boldsymbol{Z}, \boldsymbol{Z}'$ for downstream tasks;

---

## B. Dataset Statistics

We conduct experiments on 14 real-world datasets which are all publicly available with open access. The detailed statistics are presented in Table 8.

- **Cora, Citeseer** and **PubMed** (Sen et al., 2008) are well-known citation networks, with nodes representing scientific papers and edges representing citation relations. Node features are bag-of-words vectors of papers, and labels represent domains of papers.

- **Amazon-Photo** and **Amazon-Computers** (Shchur et al., 2018) are networks of co-purchase relations from Amazon. Nodes are products, edges exist when two products are frequently co-purchased. Each node with a bag-of-words feature encoding product reviews is labeled with product category.

- **Wiki-CS** (Mernyei & Cangea, 2020) is a reference network from Wikipedia references. Nodes are articles about computer science and edges are hyperlinks between the articles. Articles are labeled with 10 related subfields, and their features are average of pre-trained Glove (Pennington et al., 2014) word embeddings.

- **Squirrel** (Rozemberczki et al., 2021) is a page-page network on specific topics in Wikipedia. In this dataset, nodes represent articles from the English Wikipedia and edges reflect mutual links between them. Node features indicate the presence of particular nouns in the articles. We classify nodes into 5 classes in terms of their average monthly traffic.

- **Actor** (Tang et al., 2009) is a type of co-occurrence network based on the Microsoft Academic Graph. Nodes represent actors, and edges represent co-occurrence on the same Wikipedia page. Node features correspond to some keywords in the Wikipedia pages. We classify the nodes into 5 categories in terms of words of actor's Wikipedia.

- **Cornell, Texas** and **Wisconsin** (Pei et al., 2020) are subdatasets of WebKB, a webpage dataset from the computer science departments of Carnegie Mellon University. Nodes represent web pages and edges represent hyperlinks between them. Node features are the bag-of-words representation of web pages. We classify the web pages into 5 categories including student, project, course, staff and faculty.

- **Ogbn-Arxiv** (Hu et al., 2020) is a citation network between all Computer Science arXiv papers. Each node is an arXiv paper, and each directed edge indicates the citations between papers. Each paper with a 128-dimensional feature vector, obtained by averaging the word embeddings in its title and abstract computed with the skip-gram model (Mikolov et al., 2013).

- **Ogbn-Products** (Hu et al., 2020) is an Amazon product co-purchasing network. Nodes represent products in Amazon, and edges indicate that two products are purchased together. Node features are product descriptions, generated by Principal Component Analysis on bag-of-words features with 100 dimension.

- **Ogbn-Papers100M** (Hu et al., 2020) is a citation graph of 111 million papers. Its graph structure and node features are constructed in the same way as Ogbn-Arxiv. Approximately 1.5 million arXiv papers are manually labeled with arXiv's subject areas. Overall, this dataset is orders-of-magnitude larger than any existing node classification datasets.

*Table 8.* The statistics of 14 benchmark datasets.

| Dataset | Property | Nodes | Edges | Features | Labels |
|---|---|---|---|---|---|
| Cora | homophilic | 2,708 | 5,429 | 1,433 | 7 |
| Citeseer | homophilic | 3,327 | 4,732 | 3,703 | 6 |
| Pubmed | homophilic | 19,717 | 44,338 | 500 | 3 |
| Amazon-Photo | homophilic | 7,650 | 119,081 | 745 | 8 |
| Amazon-Computers | homophilic | 13,752 | 245,861 | 767 | 10 |
| Wiki-CS | homophilic | 11,701 | 216,123 | 300 | 10 |
| Squirrel | heterophilic | 5,201 | 217,073 | 2,089 | 5 |
| Actor | heterophilic | 7,600 | 33,544 | 931 | 5 |
| Cornell | heterophilic | 183 | 295 | 1,703 | 5 |
| Texas | heterophilic | 183 | 309 | 1,703 | 5 |
| Wisconsin | heterophilic | 251 | 499 | 1,703 | 5 |
| Ogbn-Arxiv | homophilic | 169,343 | 1,166,243 | 128 | 40 |
| Ogbn-Products | homophilic | 2,449,029 | 61,859,140 | 100 | 47 |
| Ogbn-Papers100M | homophilic | 111,059,956 | 1,615,685,872 | 128 | 172 |

## C. Theoretical Proofs

**Theorem 1.** *For the SBM graph $\mathcal{G} = (\boldsymbol{X}, \boldsymbol{A})$, we construct an even partition $\mathcal{K} = \{\mathcal{P}_1, \mathcal{P}_2, \ldots, \mathcal{P}_K\}$, with the corresponding community assignment matrix $\mathcal{C} \in \{0,1\}^{N \times N'}$. The partition residual matrix is defined as $\boldsymbol{R} = \boldsymbol{A} - \mathcal{C}\mathcal{C}^\top$. Let the encoder $f(\cdot; \boldsymbol{W})$ be a GCN with $l$ layers, and let the total weight matrix be $\boldsymbol{W}_{total} = \boldsymbol{W}^{(l)}\boldsymbol{W}^{(l-1)} \cdots \boldsymbol{W}^{(1)}$. Assume that node features within each community $\mathcal{P}_i$ satisfy a local smoothness condition such that $\max_{v_j \in \mathcal{P}_i} \|\boldsymbol{x}_j - \boldsymbol{z}'_i\|_2 \leq \epsilon, \boldsymbol{z}'_i = f(\frac{1}{|\mathcal{P}_i|} \sum_{v_j \in \mathcal{P}_i} \boldsymbol{x}_j; \boldsymbol{W})$, where $\boldsymbol{z}'_i$ denotes the mean feature vector of nodes in community $\mathcal{P}_i$. Then, the difference between the original node-level contrastive loss $\mathcal{L}_{node}(\boldsymbol{W})$ and the community anchor-level contrastive loss $\mathcal{L}_{com-node}(\boldsymbol{W})$ is bounded by*

$$|\mathcal{L}_{node}(\boldsymbol{W}) - \mathcal{L}_{com-node}(\boldsymbol{W})| \leq C_1 \cdot \|\boldsymbol{R}\|_F \cdot S_X \cdot \|\boldsymbol{W}_{total}\|_2 + C_2 \cdot \epsilon \cdot \|\boldsymbol{W}_{total}\|_2$$

*where $S_X = \max_j \|\boldsymbol{x}_j\|_2$ is the maximum norm of node features; $C_1, C_2$ are constants related to the graph size and the temperature parameter $\tau$.*

*Proof.* We begin with the loss difference decomposition using the triangle inequality

$$|\mathcal{L}_{node} - \mathcal{L}_{com-node}| \leq \underbrace{|\mathcal{L}_{node} - \mathcal{L}_{com}|}_{\Delta_{\text{split}}} + \underbrace{|\mathcal{L}_{com} - \mathcal{L}_{com-node}|}_{\Delta_{\text{smooth}}}$$

where $\Delta_{\text{split}} = |\mathcal{L}_{node} - \mathcal{L}_{com}|$ captures the partitioning error, $\Delta_{\text{smooth}} = |\mathcal{L}_{com} - \mathcal{L}_{com-node}|$ captures the intra-community feature perturbation error.

This decomposition strictly follows from the triangle inequality.

$$\Delta_{\text{split}} = |\mathcal{L}_{node}(\boldsymbol{W}) - \mathcal{L}_{com}(\boldsymbol{W})|$$

$$= \left| \sum_{i,j} \left( a_{ij} - (\mathcal{C}\mathcal{C}^\top)_{ij} \right) \cdot \ell(\boldsymbol{z}_i, \boldsymbol{z}_j) \right|$$

$$= \left| \sum_{i,j} \boldsymbol{R}_{ij} \cdot \ell(\boldsymbol{z}_i, \boldsymbol{z}_j) \right|$$

$$\leq \frac{1}{\tau} \sum_{i,j} |\boldsymbol{R}_{ij}| \cdot |\boldsymbol{z}_i^\top \boldsymbol{z}_j|$$

$$\leq \frac{1}{\tau} \sum_{i,j} |\boldsymbol{R}_{ij}| \cdot \|\boldsymbol{z}_i\|_2 \cdot \|\boldsymbol{z}_j\|_2$$

$$\leq \frac{S_Z^2}{\tau} \sum_{i,j} |\boldsymbol{R}_{ij}|$$

$$= \frac{S_Z^2}{\tau} \|\boldsymbol{R}\|_1$$

$$\leq \frac{S_Z^2 \cdot N}{\tau} \cdot \|\boldsymbol{R}\|_F$$

$$\leq \frac{\lambda_{\max}^{2l} \cdot S_X^2 \cdot \|\boldsymbol{W}_{total}\|_2^2 \cdot N}{\tau} \cdot \|\boldsymbol{R}\|_F$$

$$= C_1 \cdot \|\boldsymbol{R}\|_F \cdot S_X \cdot \|\boldsymbol{W}_{total}\|_2$$

$$\Delta_{\text{smooth}} = |\mathcal{L}_{com}(\boldsymbol{W}) - \mathcal{L}_{com-node}(\boldsymbol{W})|$$

$$= \left| \frac{1}{|\mathcal{K}|} \sum_{\mathcal{P}_i \in \mathcal{K}} \left( \ell(\boldsymbol{z}_i', \boldsymbol{z}_i') - \sum_{v_j \in \mathcal{P}_i} w_{ij}^p \cdot \ell(\boldsymbol{z}_i', \boldsymbol{z}_j) \right) \right|$$

$$\leq \frac{1}{|\mathcal{K}|} \sum_{\mathcal{P}_i \in \mathcal{K}} \sum_{v_j \in \mathcal{P}_i} w_{ij}^p \cdot |\ell(\boldsymbol{z}_i', \boldsymbol{z}_i') - \ell(\boldsymbol{z}_i', \boldsymbol{z}_j)|$$

$$\leq \frac{1}{\tau |\mathcal{K}|} \sum_{\mathcal{P}_i \in \mathcal{K}} \sum_{v_j \in \mathcal{P}_i} w_{ij}^p \cdot \|\boldsymbol{z}_i' - \boldsymbol{z}_j\|_2 \cdot \|\boldsymbol{z}_i'\|_2$$

$$\leq \frac{1}{\tau} \cdot \|\boldsymbol{z}_i\|_2 \cdot \|\boldsymbol{z}_j - \boldsymbol{z}_i'\|_2$$

$$\leq \frac{1}{\tau} \cdot \|\boldsymbol{z}_i\|_2 \cdot \lambda_{\max}^l \cdot \epsilon \cdot \|\boldsymbol{W}_{total}\|_2$$

$$\leq \frac{S_Z \cdot \lambda_{\max}^l \cdot \epsilon \cdot \|\boldsymbol{W}_{total}\|_2}{\tau}$$

$$\leq C_2 \cdot \epsilon \cdot \|\boldsymbol{W}_{total}\|_2$$

Thus, we have

$$|\mathcal{L}_{node} - \mathcal{L}_{com-node}| \leq C_1 \cdot \|\boldsymbol{R}\|_F \cdot S_X \cdot \|\boldsymbol{W}_{total}\|_2 + C_2 \cdot \epsilon \cdot \|\boldsymbol{W}_{total}\|_2$$

where

$$C_1 = \frac{N \cdot \lambda_{\max}^{2l} \cdot S_X \cdot \|\boldsymbol{W}_{total}\|_2}{\tau},$$

$$C_2 = \frac{\lambda_{\max}^{2l} \cdot S_X \cdot \|\boldsymbol{W}_{total}\|_2}{\tau}.$$

The maximum eigenvalue $\lambda_{\max}$ of the normalized adjacency matrix $\hat{A}$ satisfies $\|\hat{A}^l\|_2 = \lambda_{\max}^l$. □

## D. Hyperparameter Choices

In hyperparameter search, we adjust the number of communities $k$ and trade-off parameter $\lambda$ in CL-GCL$_{\text{METIS}}$, as well as other deep learning hyperparameters including temperature parameter $\tau$, hidden dim, learning rate, dropout and weight decay. We apply the grid search strategy to choose the optimal hyperparameters. Specifically, we search $k$ in {300, 400, 500, 600, 700} on Cora and Citeseer datasets, {2000, 2200, 2400, 2600} on Pubmed dataset, {1000, 1100, 1200, 1300, 1400, 1500} on Amazon-Photo dataset, {1400, 1500, 1600, 1700, 1800} on Amazon-Computers and Wiki-CS datasets, $\lambda$ in {0, 0.01, 0.1, 1, 10, 100}, temperature parameter in [0.1,1] with intervals of 0.1, hidden dim from {128, 256, 300, 500, 512}, learning rate from {0.0005, 0.001, 0.005, 0.01}, epochs in {100, 200, 300, 400, 500}, dropout in {0.1, 0.2, 0.3, 0.4, 0.5} and weight decay in {1e-6, 1e-5, 2e-5, 4e-5, 5e-5, 1e-4, 4e-4, 5e-4, 7e-4}. Tables 9-10 give the hyperparameters specifications for CL-GCL$_{\text{METIS}}$ on node classification and link prediction tasks.

*Table 9.* Summary of hyperparameters on node classification task.

|  | **Cora** | **Citeseer** | **Pubmed** | **Amazon-Photo** | **Amazon-Computers** | **Wiki-CS** |
|---|---|---|---|---|---|---|
| $k$ | 400 | 300 | 2000 | 1400 | 1400 | 1600 |
| $\lambda$ | 0.01 | 10 | 1 | 1 | 0.1 | 0.1 |
| Temperature $\tau$ | 0.3 | 1 | 0.4 | 0.2 | 0.7 | 0.4 |
| # Hidden dim | 500 | 500 | 300 | 512 | 300 | 500 |
| Learning rate | 0.001 | 0.01 | 0.005 | 0.0001 | 0.0001 | 0.001 |
| # Epochs | 20 | 40 | 40 | 260 | 60 | 20 |
| Dropout | 0.5 | 0.4 | 0.1 | 0.2 | 0.1 | 0.2 |
| Weight decay | 5e-4 | 2e-5 | 4e-4 | 5e-4 | 1e-4 | 1e-4 |

*Table 10.* Summary of hyperparameters on link prediction task.

|  | **Cora** | **Citeseer** | **Pubmed** | **Amazon-Photo** | **Amazon-Computers** | **Wiki-CS** |
|---|---|---|---|---|---|---|
| $k$ | 700 | 700 | 2600 | 1500 | 1800 | 1700 |
| $\lambda$ | 0.01 | 1 | 0.1 | 10 | 0.1 | 10 |
| Temperature $\tau$ | 1 | 0.2 | 0.2 | 0.3 | 0.2 | 0.2 |
| # Hidden dim | 256 | 512 | 512 | 300 | 256 | 512 |
| Learning rate | 0.0005 | 0.005 | 0.001 | 0.001 | 0.0005 | 0.001 |
| # Epochs | 20 | 40 | 200 | 280 | 460 | 380 |
| Dropout | 0.3 | 0.3 | 0.1 | 0.3 | 0.4 | 0.4 |
| Weight decay | 1e-4 | 1e-5 | 4e-5 | 1e-6 | 5e-4 | 5e-4 |

## E. More Empirical Results

In this section, we supplement more experimental results to further show the similarity histograms, give hyperparameters analysis, analyze both the effectiveness and efficiency on Ogbn-Arxiv.

### E.1. Similarity Histograms of CL-GCL$_{\text{METIS}}$

As Figure 3 demonstrates, positive and negative pairs distribution over similarity varies significantly across original node features and learned node embeddings representation spaces. These phenomena provide explanations for the challenge

of existing GCL methods. It is clear that the similarity histograms of node pairs with the same label (positive pairs) and different labels (negative pairs) overlap almost completely in the original node features representation space. Therefore, how to learn distinguishable graph representation with unlabeled data and reduce the overlapping area still remains to be effectively solved.

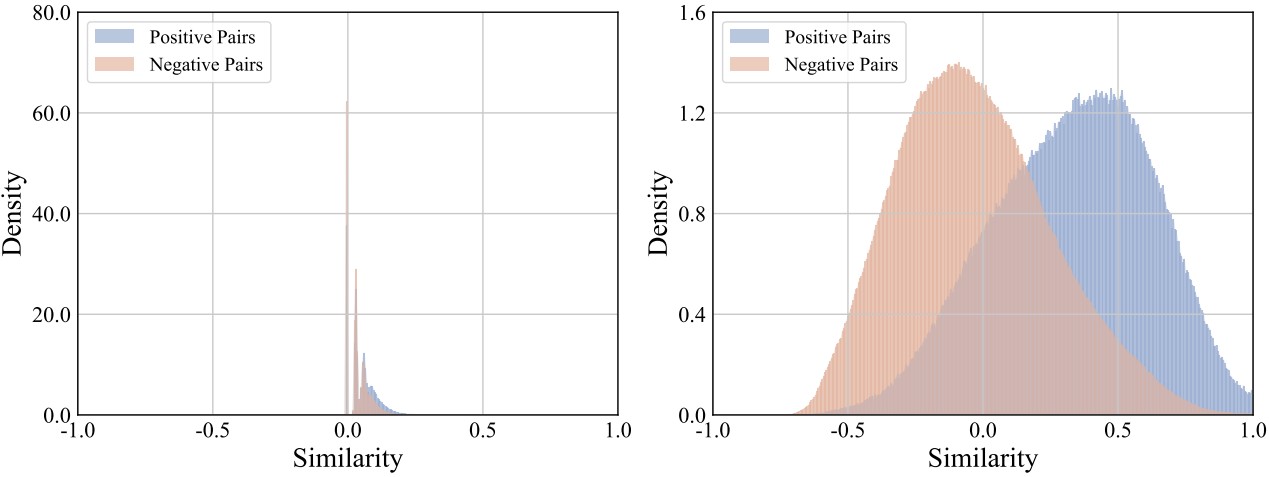

*Figure 3.* Similarity (cosine similarity) histograms of node pairs with the same label (positive pairs) and different labels (negative pairs) in the original node features vs learned node embeddings representation space of CL-GCL$_{\mathrm{METIS}}$ on Citeseer dataset.

### E.2. Hyperparameters Analysis

We analyze the impact of temperature parameter $\tau$ and hidden dim on node classification accuracy of CL-GCL$_{\mathrm{METIS}}$. Figure 4 presents the empirical results. On the whole, the optimal temperature parameter has little impact on the performance. In general, with the increase of the hidden dim, the classification performance shows an upward trend.

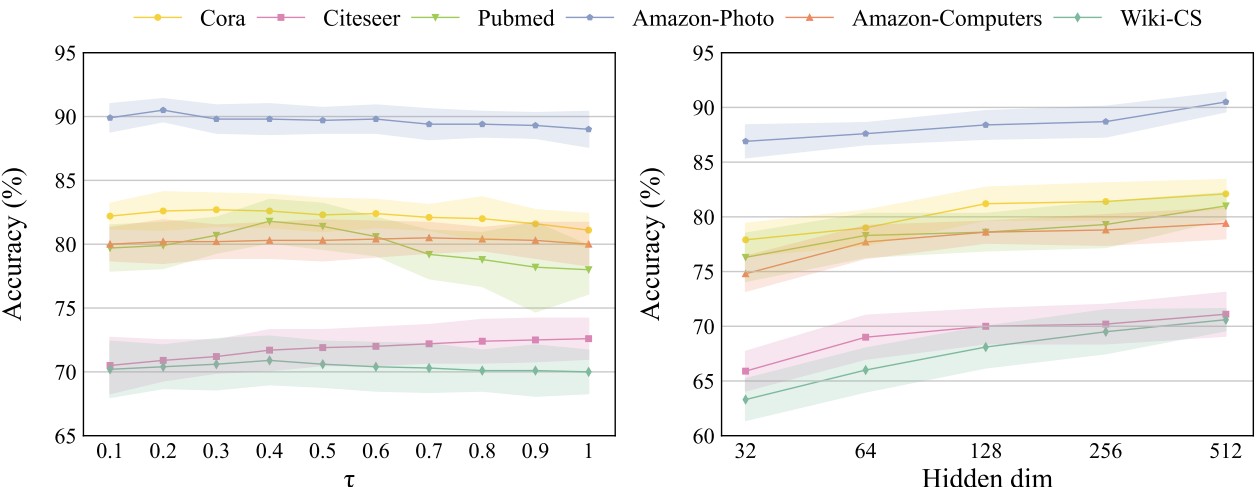

*Figure 4.* Sensitivity analysis of temperature parameter $\tau$ and hidden dim on node classification task.

The choice of $k$ is a common challenge of most graph coarsening methods such as VN, JC and GS. Tables 11-12 report the node classification accuracy and link prediction results on the Cora and Citeseer datasets with different numbers of communities $k$, respectively. The results show that CL-GCL$_{\mathrm{METIS}}$ is relatively insensitive to the choice of $k$. The performance remains stable over a wide range of values.

*Table 11.* Node classification accuracy (mean ± std) (%) with different numbers of communities $k$

| Dataset | 100 | 200 | 300 | 400 | 500 | 600 | 700 |
|---|---|---|---|---|---|---|---|
| Cora | $80.3 \pm 1.2$ | $80.5 \pm 1.0$ | $81.8 \pm 0.8$ | $82.7 \pm 1.3$ | $82.1 \pm 1.4$ | $81.9 \pm 1.4$ | $81.7 \pm 1.3$ |
| Citeseer | $70.0 \pm 1.8$ | $71.1 \pm 1.6$ | $72.6 \pm 1.6$ | $71.8 \pm 1.4$ | $71.7 \pm 1.6$ | $71.7 \pm 1.8$ | $71.5 \pm 1.2$ |

*Table 12.* Link prediction results (%) with different numbers of communities $k$

| Dataset | 100 | | 200 | | 300 | | 400 | | 500 | | 600 | | 700 | |
|---|---|---|---|---|---|---|---|---|---|---|---|---|---|---|
| | AUC | AP | AUC | AP | AUC | AP | AUC | AP | AUC | AP | AUC | AP | AUC | AP |
| Cora | 91.3 | 89.8 | 93.0 | 91.8 | 96.8 | 93.5 | 96.9 | 93.7 | 97.3 | 94.4 | 97.5 | 94.9 | 97.6 | 95.1 |
| Citeseer | 93.3 | 86.9 | 94.5 | 88.5 | 96.1 | 91.2 | 96.8 | 92.8 | 97.3 | 94.0 | 97.6 | 94.5 | 98.2 | 95.8 |

### E.3. Experiments on Large Graphs

As Tables 13-15 show, we evaluate the performance of CL-GCL on the Ogbn-Arxiv, Ogbn-Products and Ogbn-Papers100M datasets and compare it with supervised GCN (Kipf & Welling, 2017), MLP, Node2vec (Grover & Leskovec, 2016) and 5 self-supervised baselines including DGI (Velickovic et al., 2019), GRACE (Zhu et al., 2020), BGRL (Thakoor et al., 2021) and GBT (Bielak et al., 2022).

The results of the supervised and unsupervised methods come from the literature (Zheng et al., 2022). CL-GCL achieves the highest test accuracy among all methods, reaching 75.7 ± 0.3 % on the Ogbn-Products dataset, improving by 10.6% and 9.8% compared with DGI and BGRL, respectively. Additionally, CL-GCL outperforms the supervised GCN baseline on the Ogbn-Arxiv dataset, demonstrating that even without labels, contrastive learning methods can still achieve competitive performance. In terms of training efficiency, CL-GCL converges in only 20 epochs with a total training time of 105.8 seconds on the Ogbn-Arxiv dataset, significantly faster than DGI (662s), BGRL (532s), and GBT (681s), and requires 5 times fewer training epochs. Regarding memory usage, CL-GCL uses 14.97 GiB of GPU memory, comparable to GBT (15.07 GiB), and lower than BGRL (16.62 GiB) on the Ogbn-Arxiv dataset, reflecting good computational resource efficiency. In conclusion, CL-GCL demonstrates superior performance, faster convergence, and a good balance between accuracy, efficiency, and scalability while maintaining relatively low resource consumption.

*Table 13.* Node classification accuracy (mean ± std) (%), GPU memory (GiBs), time (s/epoch) and total time (s) usage on Ogbn-Arxiv. OOM indicates out-of-memory.

| Method | Test | Memory | Time | Total |
|---|---|---|---|---|
| Supervised GCN | $71.7 \pm 0.3$ | – | – | – |
| MLP | $55.5 \pm 0.2$ | – | – | – |
| Node2vec | $70.1 \pm 0.1$ | – | – | – |
| DGI (100 epos) | $70.0 \pm 0.3$ | 14.61 | 6.62 | 662.00 |
| GRACE (10k epos) | $71.5 \pm 0.1$ | OOM | / | / |
| BGRL (100 epos) | $70.1 \pm 0.3$ | 16.62 | 5.32 | 532.00 |
| GBT (100 epos) | $70.1 \pm 0.1$ | 15.07 | 6.81 | 681.00 |
| CL-GCL (20 epos) | $71.8 \pm 0.2$ | 14.97 | 5.29 | 105.80 |

*Table 14.* Node classification accuracy (mean ± std) (%), GPU memory (GiBs), time (min/epoch) and total time (min) usage on Ogbn-Products. OOM indicates out-of-memory.

| Method | Test | Memory | Time | Total |
|---|---|---|---|---|
| Supervised GCN | 75.6 ± 0.2 | – | – | – |
| MLP | 61.1 ± 0.0 | – | – | – |
| Node2vec | 68.8 ± 0.0 | – | – | – |
| DGI (20 epos) | 65.1 ± 0.1 | 22.32 | 38.43 | 768.60 |
| GRACE | – | OOM | – | – |
| BGRL (20 epos) | 65.9 ± 0.2 | 23.62 | 40.53 | 810.60 |
| GBT (20 epos) | 68.6 ± 0.1 | 22.64 | 36.74 | 734.80 |
| CL-GCL (10 epos) | 75.5 ± 0.2 | 22.53 | 45.34 | 453.40 |
| CL-GCL (15 epos) | 75.7 ± 0.3 | 22.54 | 45.65 | 684.70 |

*Table 15.* Node classification accuracy (mean ± std) (%), GPU memory (GiBs) and time usage on Ogbn-Papers100M. OOM indicates out-of-memory.

| Method | Test | Memory | Time |
|---|---|---|---|
| Supervised GCN | 66.5 ± 0.2 | – | – |
| MLP | 49.6 ± 0.3 | – | – |
| Node2vec | 58.1 ± 0.0 | – | – |
| DGI | 59.2 ± 0.4 | 22.31 | 16h27min |
| GRACE | – | OOM | – |
| BGRL (1 epo) | 62.4 ± 0.6 | 25.14 | 22h34min |
| GBT (1 epo) | 61.7 ± 0.4 | 25.73 | 20h43min |
| CL-GCL (1 epo) | 63.8 ± 0.2 | 24.68 | 18h53min |

