# OpenReview forum: "CL-GCL: Comprehensive and Lightweight Graph Contrastive Learning"
_ICML.cc/2026/Conference — ICML 2026 regular_

### Official Review · Reviewer_HVHq · 2026-03-04

**Soundness:** 3
**Presentation:** 3
**Significance:** 2
**Originality:** 3
**Overall Recommendation:** 5
**Confidence:** 3

**Summary:**

In this paper, the authors propose a community-aware graph contrastive learning framework named CL-GCL. Specially, they argue that conventional augmentation-based methods may distort high-level semantics and incur excessive computational cost due to instance-level contrast. To address these limitations, they employ graph coarsening to construct community-level anchors and design mixed community–node contrastive objectives. Extensive experiments on benchmark datasets demonstrate the performance of the proposed CL-GCL.

**Compliance With Llm Reviewing Policy:**

Affirmed.

**Final Justification:**

Most of my concerns have been solved, I have increased my score.

**Key Questions For Authors:**

Please see weaknesses.

**Limitations:**

Please see weaknesses.

**Strengths And Weaknesses:**

Strengths:

- In this paper, the authors introduce a community-aware contrastive learning framework, which provides a structured and computationally efficient framework for graph learning.
-  Theoretical analysis in section 3.5 is introduced for providing solid background for the CL-GCL.
- Extensive experiments conducted on multiple node classification and link prediction benchmarks show the performance of the proposed CL-GCL.

Weaknesses:

- The authors mention they divide the graph into community, so please consider using the Community and Erdos & Renyi datasets for conducting experiments [1].

- More recent GCL baselines need to be introduced for fair comparisons, such as [2,3,4].
- In the appendix, I find that the performance of CL-GCL is much close to Supervised GCN on the Ogbn-Arxiv dataset, is this mean that in large-scale dataset, the proposed method is equal to general GCN?
- In table 4, when employing different backbone to replace the METIS, the performance will reduced. Specially, when using GS, the model can not achive SOTA compared with baselines in table 1. So, is the performance improvement from METIS instead of the mechanism in CL-GCL?

[1] Satorras V G, Hoogeboom E, Welling M. E (n) equivariant graph neural networks[C]//International conference on machine learning. PMLR, 2021: 9323-9332.

[2] Xu Y, Huang S, Zhang H, et al. Why does dropping edges usually outperform adding edges in graph contrastive learning?[C]//Proceedings of the AAAI Conference on Artificial Intelligence. 2025, 39(20): 21824-21832.

[3] Huang Y, Zhao J, He D, et al. Does GCL need a large number of negative samples? Enhancing graph contrastive learning with effective and efficient negative sampling[C]//Proceedings of the AAAI Conference on Artificial Intelligence. 2025, 39(16): 17511-17518.

[4] Ning Z, Wang P, Qiao Z, et al. Rethinking graph contrastive learning through relative similarity preservation[C]//Proceedings of the Thirty-Fourth International Joint Conference on Artificial Intelligence. 2025: 3217-3225.

---

> ### Author Rebuttal · Authors · 2026-03-30
>
> Thank you for your detailed review. We would like to address your concerns below.
>
> **W1**: We would like to clarify two points regarding the Community and Erdős–Rényi datasets. First, the datasets referenced in the paper are not publicly available, which makes direct experimental comparison impossible. Second, the design of CL-GCL is inspired by manifold learning, involving community-aware partitioning and neighborhood sampling. Such design is more effective and efficient for capturing critical community structures in medium-scale and large-scale graphs. In contrast, for standard graph classification benchmarks, each graph is typically small in size, making community-level graph coarsening less necessary in such scenarios.
>
> **W2**: The codes of methods [2,3] are publicly available, while [4] is not. We compare with [2,3] and report the results in the tables below.
>
> #### Node classification accuracy (%) (mean ± std) on 6 datasets
> |Method|Cora|Citeseer|Pubmed|Amazon-Photo|Amazon-Computers|Wiki-CS|
> |-|-|-|-|-|-|-|
> |E2Neg|81.2±1.4|71.0±1.4|79.9±1.7|90.2±1.5|80.1±1.2|70.5±1.6|
> |EPAGCL|82.5±1.3|71.1±1.5|81.5±1.1|89.9±1.2|80.2±1.4|70.3±1.5|
> |$CL\text{-}GCL_{METIS}$|**82.7±1.3**|**72.6±1.6**|**81.8±1.7**|**90.5±0.9**|**80.5±1.2**|**70.9±1.9**|
> #### Link prediction results (%) on 6 datasets
> |Method|Cora|Citeseer|Pubmed|Amazon-Photo|Amazon-Computers|Wiki-CS|
> |---------|-----------------|-----------------|-----------------|-----------------|-----------------|-----------------|
> ||AUC&nbsp;&nbsp;&nbsp;&nbsp;AP|AUC&nbsp;&nbsp;&nbsp;&nbsp;AP|AUC&nbsp;&nbsp;&nbsp;&nbsp;AP|AUC&nbsp;&nbsp;&nbsp;&nbsp;AP|AUC&nbsp;&nbsp;&nbsp;&nbsp;AP|AUC&nbsp;&nbsp;&nbsp;&nbsp;AP|
> |E2Neg|97.1&nbsp;&nbsp;&nbsp;&nbsp;94.7|**98.3**&nbsp;&nbsp;&nbsp;&nbsp;**97.2**|96.9&nbsp;&nbsp;&nbsp;&nbsp;93.1|**95.9**&nbsp;&nbsp;&nbsp;&nbsp;91.8|**94.8**&nbsp;&nbsp;&nbsp;&nbsp;90.3|93.7&nbsp;&nbsp;&nbsp;&nbsp;88.6|
> |EPAGCL|**97.7**&nbsp;&nbsp;&nbsp;&nbsp;**95.6**|**98.3**&nbsp;&nbsp;&nbsp;&nbsp;97.1|96.8&nbsp;&nbsp;&nbsp;&nbsp;92.3|92.8&nbsp;&nbsp;&nbsp;&nbsp;86.3|92.6&nbsp;&nbsp;&nbsp;&nbsp;87.0|93.6&nbsp;&nbsp;&nbsp;&nbsp;87.7|
> |$CL\text{-}GCL_{METIS}$|97.6&nbsp;&nbsp;&nbsp;&nbsp;95.1|98.2&nbsp;&nbsp;&nbsp;&nbsp;95.8|**98.6**&nbsp;&nbsp;&nbsp;&nbsp;**97.0**|94.6&nbsp;&nbsp;&nbsp;&nbsp;**93.7**|94.5&nbsp;&nbsp;&nbsp;&nbsp;**93.9**|**93.8**&nbsp;&nbsp;&nbsp;&nbsp;**88.8**|
>
> [1] Satorras V G, Hoogeboom E, Welling M. E (n) equivariant graph neural networks[C]//International conference on machine learning. PMLR, 2021: 9323-9332.
>
> [2] Xu Y, Huang S, Zhang H, et al. Why does dropping edges usually outperform adding edges in graph contrastive learning?[C]//Proceedings of the AAAI Conference on Artificial Intelligence. 2025, 39(20): 21824-21832.
>
> [3] Huang Y, Zhao J, He D, et al. Does GCL need a large number of negative samples? Enhancing graph contrastive learning with effective and efficient negative sampling[C]//Proceedings of the AAAI Conference on Artificial Intelligence. 2025, 39(16): 17511-17518.
>
> [4] Ning Z, Wang P, Qiao Z, et al. Rethinking graph contrastive learning through relative similarity preservation[C]//Proceedings of the Thirty-Fourth International Joint Conference on Artificial Intelligence. 2025: 3217-3225.
>
> **W3**: We thank the reviewer for the observation. It is important to note that CL-GCL is an unsupervised contrastive learning method, whereas GCN is a supervised model that uses node labels.
>
> Performance achievement of CL-GCL is comparable to that of fully supervised GCN on Ogbn-Arxiv is highly significant, as it demonstrates that CL-GCL can learn high-quality representations without labels, highlighting its effectiveness in large-scale and label-scarce scenarios.
>
> **W4**: Indeed, when we use alternative backbone methods, the performance of CL-GCL shows a slight decrease. The primary purpose of this experiment is to investigate the impact of graph coarsening on the performance of CL-GCL.
>
> Specifically, we exploit graph coarsening to preserve structural semantics through community-level representations and
> manifold learning to capture local geometric relations without costly pairwise distance computations. This design naturally aligns with the neighborhood aggregation principle of Graph Convolutional Networks, thereby enhancing structural consistency while eliminating negative sampling bias.
>
> Therefore, the effectiveness of CL-GCL does not primarily depend on METIS.

---

> > ### Author Rebuttal · Reviewer_HVHq · 2026-04-01
> >
> > Thanks for your rebuttal. Most of my concerns have been addressed. I will adjust my score.

---

> > > ### Author Response · Authors · 2026-04-02
> > >
> > > We greatly appreciate your insightful feedback and positive recognition. We sincerely thank your valuable comments and suggestions.

---

### Official Review · Reviewer_GMcm · 2026-03-12

**Soundness:** 4
**Presentation:** 3
**Significance:** 4
**Originality:** 3
**Overall Recommendation:** 5
**Confidence:** 5

**Summary:**

Existing graph contrastive learning methods face the challenges of computational complexity and semantic distortion. To alleviate such limitations, this paper proposes a novel Comprehensive and Lightweight Graph Contrastive Learning (CL-GCL) framework, which uses graph coarsening to preserve critical structures and manifold learning to achieve global nonlinear structure recovery. Both theoretical analysis and empirical results demonstrate the effectiveness and efficiency of CL-GCL compared with state-of-the-art methods.

**Compliance With Llm Reviewing Policy:**

Affirmed.

**Final Justification:**

I keep my score unchanged.

**Key Questions For Authors:**

Please refer to weaknesses.

**Limitations:**

yes

**Strengths And Weaknesses:**

Strengths：

a. The paper is well-motivated. It clearly points out the computational bottleneck of existing GCL methods. The design of graph coarsening and weighted contrastive loss both serve the goal of reducing computational complexity and improving representation quality.

b. It theoretically proves that CL-GCL approximates node-level contrastive loss under mild conditions, which offers insights into its effectiveness.

c. Extensive experiments on multiple datasets of various scales cover node classification and link prediction tasks. Comparison with a large number of baseline methods, different graph coarsening methods, and ablation studies make the experimental conclusions more reliable. The results on Ogbn-Arxiv further validated its potential on large-scale graphs.

Weaknesses：

a. Graph coarsening and graph clustering are two closely related techniques easily to be confused in graph data analysis. Could the author elaborate on the differences and connections between the two in detail?

b. In the past few years, there have been some works that combine graph clustering and contrastive learning such as MCGC [1] and CCGC [2]. The authors need to clearly explain the differences of the proposed strategy compared to existing clustering based GCL methods.

[1] Multi-view contrastive graph clustering. NeurIPS 2021.

[2] Cluster-Guided Contrastive Graph Clustering Network. AAAI 2023.

---

> ### Author Rebuttal · Authors · 2026-03-30
>
> We thank the reviewer for insightful comments.
>
> **W1**:  We clarify the differences and connections between graph coarsening and graph clustering as follows.
>
> Graph coarsening aims to generate a smaller graph while preserving the structure of original graph as much as possible. It is mainly used to reduce computational complexity or provide auxiliary structures for downstream tasks.  Graph clustering partitions nodes into clusters such that nodes within the same cluster are highly similar, while nodes across clusters are less similar. It is primarily used for community detection or structure analysis.
>
> **Connections**: Graph coarsening can be viewed as a coarse-grained form of clustering. Many coarsening methods implicitly perform a clustering-like partitioning of nodes during super-nodes construction, where nodes with strong structural affinity are grouped together. In this sense, the clustering result serves as an intermediate step further compressed into a coarsened graph. Therefore, both techniques use node similarity and graph topology, and follow the same underlying principle of partition.
>
> **Differences**: Graph coarsening focuses on computational efficiency and global structural properties preservation at a reduced scale, whereas graph clustering focuses more on critical community structures capturing .
>
> **W2**: While there have been recent works combining graph clustering with contrastive learning, such as MCGC and CCGC, CL-GCL differs from them in several key aspects.
>
> 1. Core methodology: MCGC and CCGC are essentially clustering-based contrastive learning methods, where positive pairs are primarily defined by clusters. In contrast, CL-GCL is fundamentally a graph contrastive learning framework, and graph coarsening serves only as an auxiliary mechanism to construct multiple positive pairs.
>
> 2. Node sampling: CL-GCL exploit graph coarsening to preserve structural semantics through community-level representations and manifold learning to capture local geometric, providing richer and multi-scale signals. By comparison, clustering-based methods typically use only intra-cluster nodes or cluster centers, making it difficult to capture local geometric structures.

---

> > ### Author Rebuttal · Reviewer_GMcm · 2026-04-03
> >
> > Thank you for the efforts, which have effectively addressed all my concerns.

---

> > > ### Author Response · Authors · 2026-04-08
> > >
> > > Thank you very much for your time and effort.

---

### Official Review · Reviewer_MAZM · 2026-03-13

**Soundness:** 2
**Presentation:** 3
**Significance:** 2
**Originality:** 2
**Overall Recommendation:** 4
**Confidence:** 3

**Summary:**

This paper proposes CL-GCL, a self-supervised graph representation learning framework that tackles two main problems: random augmentations that damage graph semantics, and inefficient node pair sampling strategies. Instead of random augmentation, CL-GCL compresses the graph into communities using METIS by grouping similar nodes into super-nodes, and forms positive pairs between community centers and their member nodes. A soft weighting scheme assigns higher importance to nodes closer to their community center. A GCN encoder is applied to both the original and coarsened graphs and a weighted contrastive loss is optimized over these community-node pairs. Experiments show CL-GCL matches or outperforms current methods on node classification and link prediction, while being significantly faster and more memory efficient.

**Compliance With Llm Reviewing Policy:**

Affirmed.

**Final Justification:**

The rebuttal has satisfactorily addressed my concerns. I will increase my score to reflect this.

**Key Questions For Authors:**

Q1 The positive and negative pair weights W^p and W^n are computed before training and remain fixed throughout. As the model learns better representations during training, would it not make more sense to update these weights dynamically? Have the authors considered or experimented with updating the weights during training?

Q2 The paper uses METIS for graph coarsening which is fixed and not learned. Have the authors considered using a learnable coarsening method that could adapt during training and potentially give better community structures?

Q3 The paper claims to be scalable but only tests on Ogbn-Arxiv which is a small dataset. Can the authors provide results on bigger datasets like Ogbn-Products or Ogbn-MAG to better support this claim?

**Limitations:**

The authors should also discuss that the method is only tested on homogeneous graphs and may not work directly on heterogeneous graphs.

**Strengths And Weaknesses:**

$\textbf{Strengths}$
1. The paper removes each component one by one and tests on all 6 datasets. This clearly shows that both positive weights W^p and negative weights W^n actually help the performance

2. Model compresses the graph into communities to create a second view and assigns different importance weights to nodes based on how close they are to their community center. This combination is new and makes more sense than random augmentation.

3. The method is much faster and uses much less memory than all other methods on all 6 datasets, up to 614x faster and 98% less memory.

4. The paper tests 4 different graph coarsening methods and all give similar results. This shows the method is not dependent on one specific coarsening algorithm.

$\textbf{Weaknesses}$

1. The paper claims to be scalable but Ogbn-Arxiv which they test on is actually a small dataset according to the official OGB benchmark. Medium and large datasets like Ogbn-Products (2.4M nodes) and Ogbn-Papers100M (111M nodes) were never tested. So the scalability claim is not fully proven.

2. The paper only evaluates on node classification and link prediction tasks but missed graph classification.

3. The positive and negative pair weights are fixed before training and do not improve as the model learns

4. The graph coarsening is also fixed before training and does not improve as the model learns.

---

> ### Author Rebuttal · Authors · 2026-03-30
>
> Thank you for your detailed review. We would like to address your concerns below.
>
> **W1**: To test the scalability of CL-GCL more comprehensively, we conduct experiments on larger-scale datasets including Ogbn-Products (2.4M nodes), Ogbn-Papers100M (111M nodes) and report the experimental results in the tables below. Experimental results demonstrate the excellent performance of CL-GCL compared with other baseline methods.
> #### Node classification accuracy (%) (mean ± std), GPU memory (GiBs), time (min/epoch) and total time (min) usage on Ogbn-Products
> |Method|Test|Memory|Time|Total|
> |-|-|-|-|-|
> |Supervised SGC|75.6±0.2|-|-|-|
> |MLP|61.1±0.0|-|-|-|
> |Node2Vec|68.8±0.0|-|-|-|
> |DGI (20epo)|65.1±0.1|22.32|38.43|768.60|
> |GRACE|-|OOM|-|-|
> |BGRL (20epo)|65.9±0.2|23.62|40.53|810.60|
> |GBT (20epo)|68.6±0.1|22.64|36.74|734.80|
> |**CL-GCL** (10epo)|75.5±0.2|22.53|45.34|453.40|
> |**CL-GCL** (15epo)|75.7±0.3|22.54|45.65|684.70|
>
> #### Node classification accuracy (%) (mean ± std), GPU memory (GiBs) and time usage on Ogbn-Papers100M
> |Method|Test|Memory|Time|
> |-|-|-|-|
> |Supervised SGC|66.5±0.2|-|-|
> |MLP|49.6±0.3|-|-|
> |Node2Vec|58.1±0.0|-|-|
> |DGI|59.2±0.4|22.31|16h27min|
> |GRACE|-|OOM|-|
> |BGRL (1epo)|62.4±0.6|25.14|22h34min|
> |GBT (1epo)|61.7±0.4|25.73|20h43min|
> |**CL-GCL** (1epo)|63.8±0.2|24.68|18h53min|
>
> **W2**: Inspired by manifold learning, CL-GCL adopts community-aware partitioning and neighborhood sampling. Such design is more beneficial to capturing critical community structures of medium-scale and large-scale graphs effectively and efficiently. For standard graph classification benchmarks, each graph is typically small. In such cases, community-level graph coarsening is not particularly necessary.
>
> **W3**: In CL-GCL, the positive and negative pair weights are precomputed and fixed before training, not dynamically updated during training. This design has several advantages.
>
> 1. Stability enhancement
>
> Computing weights in raw feature space can serve as prior information for the input data. In contrast, computing weights after GCN processing would risk instability, as updates to GCN parameters could disrupt the construction of positive pairs. Furthermore, positive pair sampling should depend on the graph structure and node features themselves, rather than the encoder’s initial state.
>
> 2. Efficiency improvement
>
> Traditional GCL methods require pairwise similarity computations in the embedding space, whereas CL-GCL computes weights with a closed-form solution, reducing computational costs significantly.
>
> In summary, our design is inspired by manifold learning, ensuring stable positive pair construction and computational efficiency. The experimental results validate the effectiveness, demonstrating both the theoretical and practical soundness.
>
> **W4**: In CL-GCL, graph coarsening is a preprocessing stage that remains fixed during training. It is a reasonable design.
>
> First, fixed coarsened graphs ensure stable sampling and consistent representations,  as we perform node sampling on the coarsened graph and original graph.
>
> Second, METIS graph coarsening method only uses graph topology to partition communities. In such cases, dynamic update may introduce extra computational costs and thus not necessary.
>
> Overall, fixing graph coarsening design is beneficial to both stable sampling, consistent representations and high efficiency.
>
> **Q1**: Please see **W3**. We have also conducted such experiments. The results show that dynamically updating  weights increases computational cost significantly, while the performance does not show clear improvement.
>
> **Q2**: Please see **W4**.
>
> **Q3**: Please see **W1**.
>
> **Limitations**: The results on 3 heterogeneous graphs demonstrate that CL-GCL achieves state-of-the-art performance. For more details, please refer to **W2** of Reviewer **FswP**.

---

> > ### Author Rebuttal · Reviewer_MAZM · 2026-04-02
> >
> > The rebuttal has satisfactorily addressed my concerns. I will increase my score to reflect this.

---

> > > ### Author Response · Authors · 2026-04-03
> > >
> > > Thanks a lot for your valuable comments and positive recognition.

---

### Official Review · Reviewer_FswP · 2026-03-13

**Soundness:** 3
**Presentation:** 4
**Significance:** 4
**Originality:** 3
**Overall Recommendation:** 5
**Confidence:** 4

**Summary:**

This paper proposes CL-GCL, a comprehensive and lightweight framework designed to address two key challenges in Graph Contrastive Learning (GCL): semantic corruption caused by random data augmentation and the computational overhead and inter-class entanglement resulting from traditional 1-to-1 sampling strategies. The framework leverages "graph coarsening" to preserve structural semantics at the community level and incorporates "manifold learning" to capture local geometric relationships without expensive pairwise distance computations. The authors theoretically demonstrate that this method can effectively approximate traditional contrastive losses and validate its significant improvements in accuracy, training speed, and memory efficiency across multiple benchmark datasets.

**Compliance With Llm Reviewing Policy:**

Affirmed.

**Final Justification:**

most my critical questions have been well addressed

**Key Questions For Authors:**

1. Regarding the pre-computation time of coarsening algorithms like METIS on large-scale graphs: Has this been included in the total runtime reported? And for dynamic graphs, is this overhead acceptable?
	2. Regarding the choice of the number of communities K, which appears to significantly impact performance: Is there an automated heuristic method to determine the optimal K for different datasets?
	3. When applying this method to graphs with high heterophily (where connected nodes tend to have different labels), does community-level semantic preservation remain effective?

**Limitations:**

yes

**Strengths And Weaknesses:**

Strengths:
1.	This work substantially addresses the scalability issue in GCL (achieving 1.6x to 614x speedup in training and 66% to 98% memory reduction), offering high practical value for handling large-scale industrial graph data.

2.	The technical logic of the paper is rigorous, featuring theoretical proofs (e.g., Theorem 1) that establish the connection between community-level and node-level contrastive losses. The experimental design is comprehensive, covering tasks from node classification to link prediction, and demonstrates robust performance on large-scale graphs like Ogbn-Arxiv.

3.	It creatively integrates graph coarsening with contrastive learning, introducing a novel perspective of sampling at the community level rather than the node level to avoid "semantic shift."


Weaknesses:
1.	The experiments lack a discussion and analysis of the key parameter, the number of communities K. Whether the method is sensitive to the choice of K and how to select K for better performance are open questions that need addressing.

2.	The paper primarily focuses on node classification and link prediction tasks on homogeneous undirected graphs. For heterogeneous graphs, where community definitions and structural semantics may require more complex mechanisms, the CL-GCL approach may not be directly applicable or effective. While future work mentions extending it to incremental spatio-temporal modeling and multimodal data, the paper does not discuss potential limitations concerning graph homophily levels.

3.     reconstructed weights learned based on the original features can be directly applied to the constantly changing node embeddings during the training process. The assumption that the reconstruction relationship in the feature space also holds in the embedding space requires more thorough discussion or justification.

4.  There are some recent community-based GCL works including CSGCL [1] and CI-GCL [2]. What are the differences between this work and these works?

[1] CSGCL: community-strength-enhanced graph contrastive learning. IJCAI 2023.
[2] Community-invariant graph contrastive learning. ICML 2024.

---

> ### Author Rebuttal · Authors · 2026-03-30
>
> Thank you for your detailed review. We would like to address your concerns below.
>
> **W1**: The choice of K is a common challenge of most graph coarsening methods such as VN, JC and GS. In this paper, we adopt grid search to tune the hyperparameter K. The table below reports the experimental results on the Cora and Citeseer datasets with different numbers of communities K. The results show that CL-GCL is relatively insensitive to the choice of K. The performance remains stable over a wide range of values.
> #### Node classification accuracy (%) (mean ± std) with different numbers of communities K
> |Dataset|100|200|300|400|500|600|700|
> |-|-|-|-|-|-|-|-|
> |Cora|80.3±1.2|80.5±1.0|81.8±0.8|82.7±1.3|82.1±1.4|81.9±1.4|81.7±1.3|
> |Citeseer|70.0±1.8|71.1±1.6|72.6±1.6|71.8±1.4|71.7±1.6|71.7±1.8|71.5±1.2|
> #### Link prediction results (%) (mean ± std) with different numbers of communities K
> |Dataset| Metric|100|200|300|400|500|600|700|
> |-|-|-|-|-|-|-|-|-|
> |Cora|AUC|91.3|93.0|96.8|96.9|97.3|97.5|97.6|
> ||AP|89.8|91.8|93.5|93.7|94.4|94.9|95.1|
> |Citeseer|AUC|93.3|94.5|96.1|96.8|97.3|97.6|98.2|
> ||AP|86.9|88.5|91.2|92.8|94.0|94.5|95.8|
>
> **W2**: As this paper primarily focuses on node classification and link prediction tasks on homogeneous undirected graphs, we have supplemented our study with experiments on heterogeneous graphs. We follow the data splits and baseline methods of GRASS (Universal Graph Self-Contrastive Learning, IJCAI 2025). The table below presents the node classification results on several commonly used heterogeneous graph datasets. The results show that CL-GCL achieves best performance compared with baseline methods.
> #### Node classification accuracy (%) (mean ± std) on 3 heterogeneous graph datasets
> |Method|Squirrel|Actor|Cornell|
> |-|-|-|-|
> |GraphACL|41.36±4.57|30.12±0.21|59.67±1.39|
> |SGCL|38.49±0.86|31.23±0.64|62.76±2.94|
> |GCIL|40.68±1.35|31.67±0.94|63.57±3.19|
> |GRASS|40.38±2.64|34.94±0.94|70.25±5.66|
> |$CL\text{-}GCL_{METIS}$|**42.01±3.12**|**35.45±1.51**|**70.39±6.12**|
>
> **W3**: Such design is inspired by the manifold assumption, i.e. , both the original feature space and the learned embedding space can be regarded as samples drawn from the same low-dimensional latent manifold, and the graph encoder learns a smooth mapping on this manifold.
>
> The weights are derived from local linear reconstruction in the original feature space. They capture the geometric structure of local neighborhoods on the manifold. Under the manifold smoothness assumption, such local geometric are approximately preserved under continuous mappings. Since graph encoders such as GCN achieve smooth transformations through neighborhood aggregation, they are able to maintain the consistency of local neighborhood structures in the embedding space, making these structures transferable in the embedding space.
>
> **W4**: While recent community-based GCL works such as CSGCL and CI-GCL also use community structures of graphs, CL-GCL differs from them in two key aspects.
>
> 1. Multi-positive node sampling
>
> CSGCL and CI-GCL construct only a single positive pair for each node during training. In contrast, CL-GCL constructs multiple positive pairs for each node.
>
> 2. Manifold learning inspired sampling
>
> CSGCL and CI-GCL ignore‌ the relative importance of different nodes. CL-GCL, inspired by manifold learning, adaptively assigns weights to positive pairs, and thus capturing comprehensive community structures effectively.
>
> These two improvements jointly enable CL-GCL to achieve excellent representation learning performance compared to existing methods.
>
> **Q1**: The precomputation time of graph coarsening algorithm is not included in the total runtime reported. The total runtime only includes the training time of CL-GCL. We observe that in existing GCL works, the reported training time typically also does not include additional preprocessing time.
>
> For dynamic graphs, this additional preprocessing time is manageable. We may adopt incremental update strategies, where graph coarsening can be periodically recomputed or applied only to the subgraphs with changes, rather than reprocessing the entire graph each time. Therefore, compared to the performance gains, this additional cost is negligible in most practical scenarios.
>
> **Q2**: Please see **W1**.
>
> **Q3**: Please see **W2**.

---

> > ### Author Rebuttal · Reviewer_FswP · 2026-04-04
> >
> > I sincerely thank the authors for their comprehensive rebuttal. As most my critical questions have been well addressed, I will adjust my score accordingly.

---

> > > ### Author Response · Authors · 2026-04-08
> > >
> > > Thank you for your acknowledgment and recognition. We are glad to know that our rebuttal have addressed your concerns. Your insights have helped refine our work.

---

### Decision · Program_Chairs · 2026-04-30

**Decision:**

Accept (regular)

**Comment:**

This paper proposes a comprehensive and lightweight Graph Contrastive Learning framework by exploiting graph coarsening and manifold learning‌. It possesses the property of enhancing structural consistency and approximating node-level contrastive loss. Experiments and the ablation study are sufficient to justify its statements. All four reviewers provide positive feedback by realizing the technical contributions, clear motivations, and theoretical analysis. The reviewers’ concerns are fully resolved in the rebuttal.